# LOST IN REAL-WORLD SCENARIOS: CONCRETIZATION DISRUPTS LLM LOGICAL REASONING

## ABSTRACT

Although large language models (LLMs) have attracted significant attention, recent studies reveal that even minor variations in input formulation can lead to substantial inconsistencies in reasoning outcomes, underscoring their fragility in real-world scenarios. To systematically investigate this issue, we propose a concretization framework that automatically translates clean reasoning logic into concrete contexts with challenging formulations. In this framework, two translators are trained via a dual-learning approach. The first converts formal language templates into natural language puzzles, guided by a difficulty-aware reward that promotes the exploration of harder formulations. The second translates puzzles back into templates, with isomorphism verification ensuring the consistency of underlying reasoning logic. Applying this framework, we efficiently build paired datasets of formal language templates and natural language puzzles, and observe a sharp drop in LLM reasoning performance when moving from templates to puzzles. To uncover the underlying causes, we conduct an in-depth analysis of how tokens derived from formal templates and natural language puzzles influence the final answers. This analysis reveals two primary sources of degradation: dispersed reasoning attention across non-essential tokens and conflicts introduced by alternative formulations. To address these issues, we propose a prompt-based approach that instructs LLMs to abstract reasoning logic from concrete contexts before attempting direct solutions, and a training-based approach that further strengthens LLMs' abstraction ability. Experimental results show that our methods improve LLM performance on natural language puzzles by up to 56.2%, nearly eliminating the performance loss induced by concretization.

## 1 INTRODUCTION

Since the advent of large language models (LLMs), reasoning has consistently been recognized as one of their most critical capabilities. The rise of large reasoning models has highlighted their remarkable performance across a wide range of reasoning tasks. However, studies have shown that variations in input formulation can substantially undermine the reasoning ability of LLMs. This fragility exposes a lack of robustness and presents significant challenges for adapting their reasoning performance to complex, real-world scenarios.

To systematically investigate this phenomenon, prior studies focus on identifying pairs of inputs that differ in surface formulation but preserve underlying reasoning logic. Trivial perturbations have been shown to negatively impact LLM reasoning performance on established benchmarks, for example, through rephrasing (Zhou et al., 2024), introducing typos (Gan et al., 2024), switching languages (Hu et al., 2025), extending context (Xu et al., 2025), or even inserting irrelevant statements such as "Interesting fact: cats sleep for most of their lives" (Rajeev et al., 2025). However, these methods are largely heuristic, focusing only on surface-to-surface variations, and lack deeper investigation into how LLMs model the relationship between surface formulation and underlying reasoning logic.

To address this issue, we propose a concretization framework that automatically converts abstract reasoning logic into specific contexts while exploring challenging formulations. Specifically, the translator is trained through a dual-learning approach. The first translator learns to translate a formal

language template, primarily encoding pure reasoning constraints, into a natural language puzzle, guided by a difficulty-aware reward that encourages exploration of more challenging formulations. The second translator learns to translate the natural language puzzle back into a formal language template, with an isomorphism verification applied to guarantee that the reasoning logic remains consistent with the original formal language template.

Using our concretization framework, we construct paired formal language templates and natural language puzzles across three problem types: SAT problems with only Boolean variables, CSP problems with Boolean + integer variables, and CSP problems with Boolean + integer + Abelian-group variables (problem definitions see Section 2.1). Across all settings, we observe a substantial decline in LLM reasoning performance when moving from abstract templates to their concretized formulations. As shown in Figure 1, the Qwen3-30B-A3B model (Yang et al., 2025a) suffers a 63.0% accuracy reduction on CSP with Boolean + integer variables.

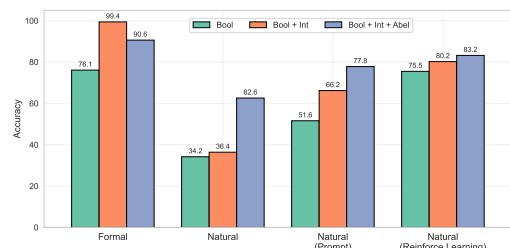

Figure 1: The performance comparison of the Qwen3-30B-A3B across formal language templates, natural language puzzles, with prompt-based method, and training-based method.

To mitigate this gap, we propose a prompt-based strategy that guides LLMs to first infer the underlying formal language template from a natural language puzzle before solving it. This method alone yields a 29.8% accuracy improvement on CSP problems with Boolean + integer variables. Building on this, we further design a training-based approach that leverages our abstraction–concretization paired data to strengthen the model's ability to abstract natural descriptions into structured formal representations, resulting in an additional 23.9% gain on CSP problems with Boolean + integer + Abelian-group variables. Notably, the abstraction-enhanced model also generalizes better to out-of-domain benchmarks, obtaining a 5.0% performance boost on PlanBench (Valmeekam et al., 2023). These results underscore that LLM reasoning is fundamentally constrained by their limited robustness in mapping concretized descriptions back to the underlying abstract structure.

To further investigate the underlying causes, we conduct a detailed analysis of how input tokens from formal language templates and natural language puzzles influence LLM predictions. Our findings reveal that LLMs often allocate disproportionate attention to reasoning-irrelevant tokens while underemphasizing reasoning-critical ones. Moreover, shifts in problem formulation lead to corresponding shifts in reasoning patterns, further exacerbating performance degradation.

To summarize, the main contributions of this paper are:

- We propose an isomorphism-verified, difficulty-aware concretization framework that automatically transform formal language templates into challenging natural language puzzles while preserving underlying reasoning logic, providing an efficient way to generate both abstraction-concretization analysis data and abstraction-enhanced training data.

- We conduct experiments on constructed paired abstraction-concretization data, we show that concretization formulation significantly reduces LLM reasoning performance, and we propose prompt-based and training-based abstraction-enhanced methods that effectively mitigate this performance drop.

- We conduct an in-depth analysis of why LLMs fail to model the relationship between surface formulations and underlying reasoning logic, identifying two key causes: disproportionate attention to reasoning-irrelevant tokens and the difficulty of maintaining consistent reasoning patterns across diverse formulations.

## 2 METHODOLOGY

An overview of the construction process for formal language template–natural language puzzle pairs is shown in Figure 2. The detailed designs of formal language template generation and natural language puzzle concretization are provided in Subsection 2.1 and Subsection 2.2. Furthermore,

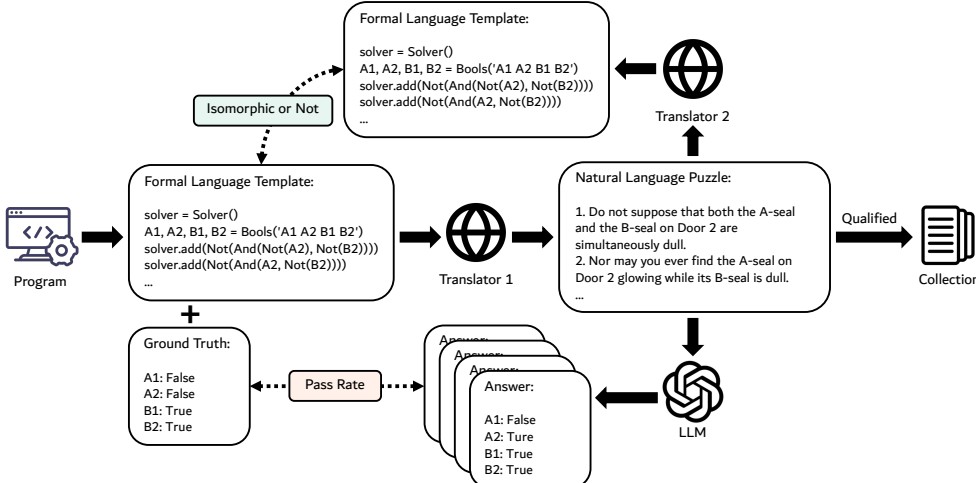

Figure 2: The construction process of a paired formal language template and natural language puzzle proceeds. First, a rule-based program generates a formal language template and its ground-truth assignment. This template is then passed to a translator, which converts it into a natural language puzzle. The puzzle is subsequently back-translated into a formal template by another translator and presented to an LLM, which produces multiple responses. A natural language puzzle is retained and collected if it passes isomorphism verification and its pass rate falls below a difficulty threshold.

Subsection 2.3 introduces our prompt-based and training-based mitigation strategies, which aim to alleviate the performance degradation of LLMs caused by concretization.

## 2.1 FORMAL LANGUAGE TEMPLATES CONSTRUCTION

**Start from the SAT Problem.** To design our formal language template, we begin with the Boolean satisfiability problem (SAT) as the target task, since it is a canonical benchmark for logical reasoning and serves as a foundational abstraction for many real-world computational problems. SAT requires finding an assignment of truth values to variables such that a given Boolean formula is satisfied. Consider the following formula in conjunctive normal form (CNF) with a $2 \times 2$ variable arrangement and four clauses:

$$F = (\neg A_1 \vee B_1) \wedge (A_1 \vee \neg B_2) \wedge (A_1 \vee B_2) \wedge (\neg A_2 \vee \neg B_2).$$

One satisfying assignment is:

$$A_1 = \text{True}, \quad A_2 = \text{False}, \quad B_1 = \text{True}, \quad B_2 = \text{False}.$$

Two SAT instances $F$ and $G$ over variable sets $\text{Var}(F)$ and $\text{Var}(G)$ are said to be *logically isomorphic* if there exists a bijection

$$\pi : \text{Var}(F) \to \text{Var}(G)$$

such that the following three conditions hold. First, $\pi$ preserves literal polarity: $\pi(\neg X) = \neg \pi(X)$. Second, $\pi$ preserves clause structure: applying $\pi$ to every literal of every clause in $F$ produces exactly the multiset of clauses in $G$. Third, all Boolean connectives remain unchanged under the mapping, so the renaming preserves the syntactic form of the CNF formula. Intuitively, the two formulas encode the same logical structure up to a renaming that treats literals consistently.

**Extension to CSP with Boolean and Integer Variables.** SAT can be viewed as a special case of the Constraint Satisfaction Problem (CSP), where all variables are Boolean and all constraints are logical clauses. We extend our formal language template to a richer CSP setting in which variables may be of type `bool` or `int`, and the task becomes finding an assignment of Boolean and integer values satisfying all relational, arithmetic, and logical constraints. Consider the CSP instance with

$$x \in \{\text{True}, \text{False}\}, \quad y \in \{0, 1, 2\}, \quad z \in \{0, 1\},$$

and constraints

$$(x = \text{True} \Rightarrow y \leq 1), \qquad (y + z = 2), \qquad (\neg x \vee (z = 1)).$$

A satisfying assignment is

$$x = \text{True}, \quad y = 1, \quad z = 1.$$

Formally, a Boolean–integer CSP instance is a triple $I = (V, \text{dom}, \mathcal{C})$, where $V$ is a set of variables, $\text{dom}(v)$ assigns a domain and a type to each variable, and $\mathcal{C}$ is a set of constraints built from a fixed language of logical predicates and arithmetic operations. Two such CSP instances $I = (V, \text{dom}, \mathcal{C})$ and $I' = (V', \text{dom}', \mathcal{C}')$ are *isomorphic* if there exists a bijection $\pi : V \to V'$ satisfying three requirements. First, variable types are preserved: $\text{dom}(v)$ and $\text{dom}'(\pi(v))$ belong to the same sort, such as `bool` or `int`. Second, the mapping commutes with term formation: whenever a term uses functions such as addition, comparison, or Boolean connectives, the mapped term is obtained simply by replacing variables according to $\pi$ while leaving all operators unchanged. Third, constraint structure is preserved: applying $\pi$ to all variables appearing in a constraint of $\mathcal{C}$ yields exactly one constraint in $\mathcal{C}'$, and every constraint of $\mathcal{C}'$ arises in this way. This definition reduces to the SAT isomorphism when all variables are Boolean and all constraints are clauses.

**Extension to CSP with Boolean, Integer, and Abelian Group Variables.** We further generalize our template to CSPs whose variables may belong to Boolean domains (e.g., $\{\text{True}, \text{False}\}$), integer domains, or Abelian group domains such as $\mathbb{Z}_p$ or $\mathbb{Z}^k$. In this enriched CSP, constraints may involve group operations, congruence relations, and linear relations over Abelian group structures.

Consider the CSP:

$$x \in \{\text{True}, \text{False}\}, \qquad y \in \{0, 1, 2\}, \qquad g \in \mathbb{Z}_4,$$

with constraints:

$$(x = \text{True} \Rightarrow y < 2), \qquad g + g \equiv y \pmod{4}, \qquad (x = \text{False} \Rightarrow g \neq 1).$$

A satisfying assignment is:

$$x = \text{False}, \quad y = 2, \quad g = 3.$$

The first implication is vacuously true because $x = \text{False}$. The second constraint holds since $g + g = 3 + 3 = 6 \equiv 2 \pmod{4}$. The third holds because $x = \text{False}$ and $g = 3 \neq 1$.

Formally, such a CSP instance again takes the form $I = (V, \text{dom}, \mathcal{C})$, but $\text{dom}(v)$ may now be a Boolean set, an integer domain, or the carrier set of a fixed Abelian group. Two CSP instances with Boolean, integer, and Abelian group variables are *isomorphic* if a bijection $\pi : V \to V'$ satisfies three structural requirements. First, variable sorts and domain structures are preserved: a Boolean variable maps to a Boolean variable, an integer variable maps to an integer variable, and a group-valued variable ranging over an Abelian group $G$ maps to another variable whose domain is the same group $G$. Second, the mapping preserves term structure, meaning that group operations, arithmetic operations, and logical connectives remain unchanged while variables appearing in terms are renamed via $\pi$. Third, constraint preservation holds exactly as before: each constraint in $\mathcal{C}$ becomes a constraint in $\mathcal{C}'$ after applying $\pi$, and the set $\mathcal{C}'$ consists precisely of such images.

Introducing Abelian group variables significantly enriches the expressive power of the CSP template. Constraints can now encode group equations, homomorphic structure, and congruence relations, and the corresponding isomorphisms must preserve not only logical and arithmetic structure but also the underlying algebraic structure induced by the group domains.

## 2.2 Natural Language Puzzle Concretization

Our concretization framework adopts the standard dual learning approach (Xia et al., 2016), which consists of two training cycles involving two translators. In the first cycle, Translator 1 translates a formal language template into a natural language puzzle, while Translator 2 translates the resulting puzzle back into a formal language template. Translator 1 serves as the optimization target in this cycle. In the second cycle, Translator 2 translates a real-world puzzle into a formal language template, and Translator 1 then translates the template back into a natural language puzzle. In this cycle, Translator 2 is the optimization target. The overall training process is illustrated in Figure 3.

For Translator 1, the input is a constructed formal language template, and the output is a natural language puzzle together with variable definitions. The reward is derived from two components: (i) the pass rate of an answer model on the generated natural language puzzle, and (ii) the isomorphism decision between the original formal language template and the back-translated template produced by Translator 2. For Translator 2, the input is a real-world puzzle, and the output is a formal language template along with variable definitions. Its reward combines (i) a format check on the generated formal language template and (ii) the similarity between the original real-world puzzle and the natural language puzzle generated by Translator 1.

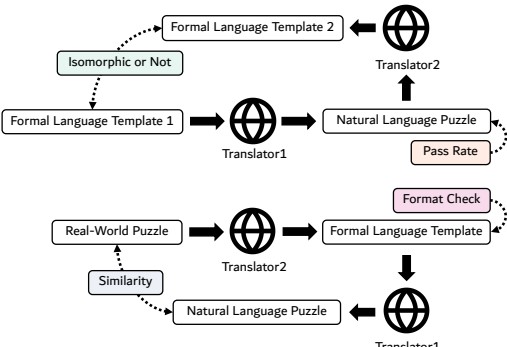

Figure 3: The training process for the natural language translator.

Through iterative training, our dual-learning framework converges toward a state where formal language templates can be automatically translated into natural language puzzles that are both challenging in formulation and logically consistent with the original formal representation.

## 2.3 MITIGATE STRATEGY

To mitigate the reasoning performance gap of LLMs when transitioning from formal language templates to natural language puzzles, we propose a prompt-based method that encourages the model to extract the underlying reasoning logic before solving the task. Specifically, the solving model is first prompted to translate the natural language puzzle into a formal language template, and then to solve this formal representation in a second step to derive the final answer. To address the tendency of reasoning models to deviate from instructions, the prompt requires the LLM to explicitly output the reconstructed formal language template. The full prompt is provided in Appendix B.

To further strengthen the model's ability to perform such abstraction, we introduce a complementary training-based method. Similar to Translator 2, the solving model is trained via reinforcement learning to translate natural language puzzles back into their corresponding formal language templates. The model takes a natural language puzzle generated by our concretization framework as input, and receives a reward based solely on whether its output formal language template is isomorphic to the original one. This objective encourages the solving model to reliably map surface formulations to their underlying reasoning logic.

## 3 EMPIRICAL RESULTS

Using the concretization framework described in Section 2, we efficiently construct paired datasets of formal language templates and natural language puzzles that preserve consistent reasoning logic while introducing more challenging surface formulations. In this work, we focus on three problem types: SAT problems with only Boolean variables, CSP problems with Boolean and integer variables, and CSP problems with Boolean, integer, and Abelian-group variables. For SAT problems, we adopt variable-size settings of $3 \times 3$, $3 \times 5$, and $5 \times 5$. For SAT and CSP problems with Boolean and integer variables, we collect 500 puzzles that satisfy a difficulty threshold defined as a pass rate below 8/16 rollouts when solved by Qwen3-30B-A3B. For CSP problems involving Boolean, integer, and Abelian-group variables, we similarly collect 500 puzzles that meet a difficulty threshold defined using GPT-oss-120B, again requiring a pass rate below 8/16 rollouts.

## 3.1 PERFORMANCE DEGRADATION AFTER CONCRETIZATION

We report the accuracy of state-of-the-art reasoning LLMs on both the original formal language templates and their corresponding natural language puzzles in Table 1. As shown, nearly all models achieve high accuracy on the formal templates but experience substantial performance drops after translation into natural language. The largest gap appears in the SAT problem with $3 \times 3$ variables

| Model | Method | Bool | | | | | | + Int | | + Abel | |
|---|---|---|---|---|---|---|---|---|---|---|---|
| | | 3 × 3 | | 3 × 5 | | 5 × 5 | | | | | |
| | | FL | NL | FL | NL | FL | NL | FL | NL | FL | NL |
| Qwen3-30B-A3B | Orig. | 97.6 | 31.6 | 89.4 | 29.8 | 41.4 | 23.2 | 99.4 | 36.4 | 90.6 | 62.6 |
| | Prom. | - | 65.6 (+34.0) | - | 60.2 (+30.4) | - | 29.2 (+6.0) | - | 66.2 (+29.8) | - | 77.8 (+15.2) |
| | RL | - | 87.8 (+56.2) | - | 84.4 (+54.6) | - | 53.4 (+30.2) | - | 80.2 (+43.8) | - | 83.2 (+21.0) |
| GPT-oss-20B | Orig. | 85.8 | 74.2 | 62.0 | 49.2 | 20.2 | 13.0 | 97.8 | 70.2 | 84.4 | 47.8 |
| | Prom. | - | 81.0 (+6.8) | - | 59.4 (+10.2) | - | 17.8 (+4.8) | - | 80.4 (+10.2) | - | 54.2 (+6.4) |
| | RL | - | 86.4 (+12.2) | - | 74.2 (+25.0) | - | 24.2 (+11.2) | - | 83.8 (+13.6) | - | 72.8 (+25.0) |
| Deepseek-R1 | Orig. | 99.8 | 73.0 | 99.0 | 61.6 | 92.6 | 71.2 | 100 | 83.8 | 97.8 | 63.6 |
| | Prom. | - | 92.2 (+19.2) | - | 81.0 (+19.4) | - | 76.4 (+5.2) | - | 88.4 (+4.6) | - | 71.8 (+8.2) |
| Gemini-2.5-Pro | Orig. | 99.2 | 80.2 | 98.8 | 74.0 | 86.6 | 80.2 | 100 | 82.2 | 99.2 | 66.8 |
| | Prom. | - | 89.4 (+9.2) | - | 78.8 (+4.8) | - | 68.6 (-11.6) | - | 87.4 (+5.2) | - | 76.0 (+9.2) |
| GPT-o3 | Orig. | 99.4 | 97.0 | 99.8 | 97.8 | 99.4 | 98.8 | 100 | 87.0 | 99.8 | 72.4 |
| | Prom. | - | 98.0 (+1.0) | - | 99.6 (+1.8) | - | 99.0 (+0.2) | - | 90.4 (+3.4) | - | 83.6 (+11.2) |

Table 1: The accuracy of LLMs on our generated abstraction–concretization paired dataset before (Orig.), after introducing the intermediate prompt-based step (Prom.), and after abstraction-enhanced reinforcement learning (RL).

using Qwen3-30B-A3B as the solving model, where accuracy decreases by 66%. Even leading closed-source models such as Gemini-2.5-Pro (Comanici et al., 2025) and GPT-o3 also show notable declines of 32.4% and 27.3%, respectively, on CSP problems involving Boolean, integer, and Abelian-group variables.

Meanwhile, although all LLMs are affected by the natural language formulation introduced through concretization, their robustness varies across settings. For the SAT problems and the CSP problems with Boolean and integer variables, where Qwen3-30B-A3B is used to define the difficulty threshold, GPT-oss-20B exhibits noticeably better robustness, with at most a 27.6% performance drop compared to Qwen3-30B-A3B's maximum drop of 66%. In contrast, for the CSP problems involving Boolean, integer, and Abelian-group variables, where the difficulty threshold is defined by GPT-oss-120B, Qwen3-30B-A3B demonstrates better robustness, showing a 30% performance drop compared to GPT-oss-20B's 36.6%.

## 3.2 PERFORMANCE MITIGATED AFTER ABSTRACTION

As shown in Table 1, introducing the prompt-based reasoning-logic abstraction step leads to clear performance improvements for most LLMs across all three types of puzzles. Notably, Qwen3-30B-A3B achieves a 34% accuracy increase on the SAT problem with 3 × 3 variables, and even GPT-o3 improves by 11.2% on the CSP problems involving Boolean, integer, and Abelian-group variables. Furthermore, when the reasoning-logic abstraction ability is strengthened through our training-based approach, the performance of Qwen3-30B-A3B and GPT-oss-20B improves even further, approaching their respective performance levels on the original formal language templates. Remarkably, Qwen3-30B-A3B on the SAT problem with 3 × 3 variables, as well as GPT-oss-20B across all SAT variable settings, even surpass their accuracy on the formal templates. This indicates that abstraction-enhanced training can reduce the dispersion of LLM reasoning across symbolic and natural-language formulations, enabling models to reason more consistently and effectively.

Another outlier is Gemini-2.5-Pro, which exhibits an 11.6% decrease in accuracy on the SAT problem with 5 × 5 variables when using the prompt-based method. A closer inspection of its outputs shows that Gemini-2.5-Pro often produces the final answer immediately after the translation step, neglecting the deeper symbolic reasoning required. This suggests that effective reasoning-logic abstraction must be paired with the ability to sustain coherent reasoning over the longer output sequences introduced by this process.

## 3.3 PERFORMANCE ENHANCEMENT ON OUT-OF-DOMAIN BENCHMARKS

We further evaluate the abstraction-enhanced model on several publicly available benchmarks from prior work (Gan et al., 2024; Rajeev et al., 2025; Valmeekam et al., 2023; Zheng et al., 2024). For PlanBench, we focus specifically on the plan-generation task. As shown in Table 2, abstraction-

| Model | Method | Typographical | | CatAttack | | Natural-Plan | | | Planbench |
|---|---|---|---|---|---|---|---|---|---|
| | | Original | Edited | Original | Edited | Calendar | Meeting | Trip | |
| **Qwen3-30B-A3B** | Orig. | 90.47 | 86.87 | 96.50 | 94.50 | 84.80 | 12.30 | 3.75 | 68.20 |
| | Prom. | 90.75 (+0.28) | 87.00 (+0.13) | 96.16 (-0.34) | 95.83 (+1.33) | 85.20 (+0.40) | 12.80 (+0.05) | 4.44 (+0.69) | 70.2 (+2.00) |
| | RL | 91.18 (+0.71) | 87.50 (+0.63) | 96.50 (+0.00) | 96.00 (+1.50) | 86.20 (+1.40) | 14.10 (+1.80) | 4.94 (+1.19) | 73.20 (+5.00) |
| **GPT-oss-20B** | Orig. | 79.18 | 70.81 | 63.00 | 61.00 | 83.90 | 4.00 | 0.00 | 47.40 |
| | Prom. | 79.81 (+0.63) | 73.32 (+2.51) | 66.67 (+3.67) | 65.16 (+4.16) | 84.80 (+0.90) | 5.8 (+1.80) | 0.00 (+0.00) | 43.20 (-4.20) |
| | RL | 82.41 (+3.23) | 76.11 (+5.3) | 70.60 (+7.60) | 70.00 (+9.00) | 85.70 (+1.80) | 9.60 (+5.60) | 0.06 (+0.06) | 55.60 (+8.20) |

Table 2: The reasoning performance of Qwen3-30B-A3B and GPT-oss-20B on public benchmarks, before (Orig.), and after prompt-based (Prom.), and training-based (RL) abstraction-enhancement.

enhanced training substantially improves LLM robustness to perturbations such as injected typos and irrelevant statements. Moreover, both models also achieve higher performance on real-world planning tasks, with the abstraction-enhanced GPT-oss-20B showing an 8.2% improvement on Plan-Bench. These results highlight concretization-based training as an effective strategy for enhancing the robustness and real-world applicability of LLM reasoning.

## 4 ANALYSIS

### 4.1 INPUT FORMULATION LEADS TO MISUNDERSTANDING

Curious about the types of errors introduced by natural language, we analyze the responses of the Qwen3-30B-A3B model on the first 100 formal language templates and natural language puzzles for each size. The errors made by the model are categorized into three types:

- Constraint Misunderstandings: The model misinterprets the natural language description, leading to incorrect constraints. For example, in one puzzle about Adam, the definition states that both $B1$ and $B4$ represent "the battery is fully charged." However, during reasoning the model assigned them different truth values, thereby generating a result directly contradictory to the definition.

- Solving Failure: The model generates assignments that conflict with the given constraints. For instance, in a narrative puzzle set at night, the constraints required that $C3$ and $C1$ could not both be false. Yet, in its final solution, the model set $C3 =$ False and $C1 =$ False, resulting in a direct conflict with the constraint.

- Formatting Errors: The model fails to follow the required output format.

Figure 4 illustrates the increase in errors made by the Qwen3-30B-A3B model on natural language puzzles compared to formal language puzzles. As the number of variables in the puzzles grows, we observe that the frequency of Constraint Misunderstandings rises only slightly, whereas the frequency of Solving Failures increases more substantially. This pattern suggests that as the difficulty of symbolic reasoning intensifies, the model's reasoning becomes less robust. Consequently, even minor perturbations in natural language, though not genuine misunderstandings for LLMs, are more likely to disrupt their reasoning process.

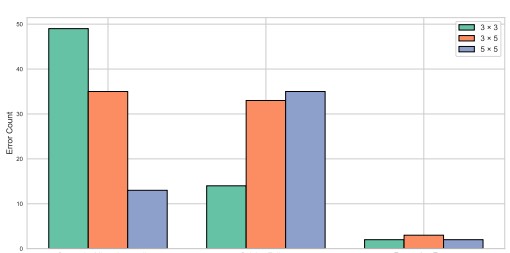

Figure 4: The increased error counts of the Qwen3-30B-A3B (puzzles minus templates).

### 4.2 REASONING ATTENTION DISPERSED ACROSS NON-REASONING TOKENS

To investigate why different prompt formulations yield divergent predictions, we measure the causal sensitivity of each input token using Grad × Input influence scores on the Qwen3-30B-A3B model. The objective function $J$ is defined as the total log-likelihood of the gold answer sequence, computed by summing the negative cross-entropy loss across the answer span. Gradients are enabled only for

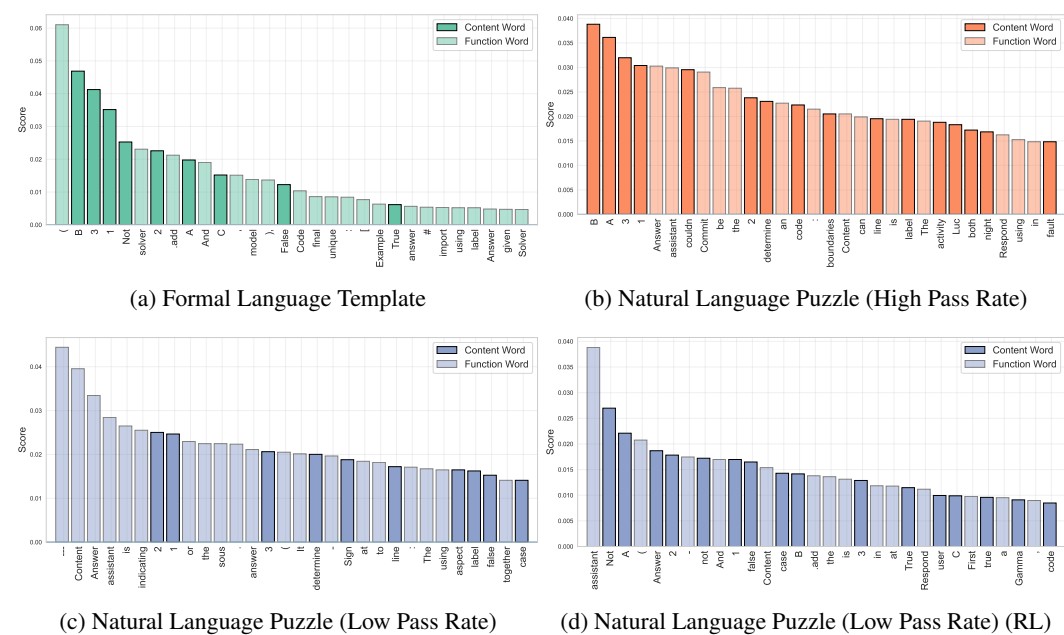

(a) Formal Language Template      (b) Natural Language Puzzle (High Pass Rate)

(c) Natural Language Puzzle (Low Pass Rate)      (d) Natural Language Puzzle (Low Pass Rate) (RL)

Figure 5: Top 30 tokens with the highest Grad × Input influence scores.

the prompt embeddings, and we backpropagate through $J$ to estimate token-level contributions. For each prompt token $i$, the Grad × Input influence score is defined as

$$\text{saliency}_i = \sum_{d=1}^{D} \frac{\partial J}{\partial e_{i,d}} \, e_{i,d}, \tag{1}$$

where $\mathbf{e}_i = (e_{i,1}, \ldots, e_{i,D}) \in \mathbb{R}^D$ denotes the embedding vector of token $i$, $\frac{\partial J}{\partial e_{i,d}}$ is the gradient of $J$ with respect to the $d$-th component of $\mathbf{e}_i$, and $D$ is the embedding dimension. Finally, we apply L1 normalization across tokens to ensure comparability.

Figure 5 presents the top 30 tokens with the highest Grad × Input influence scores. As shown, in both the formal language template and the high-pass-rate natural language puzzle, the most influential tokens are typically content words, such as negation terms (e.g., not, couldn't), variable names (e.g., B3), or semantically meaningful nouns (e.g., boundaries). By contrast, in the low-pass-rate natural language puzzle, the tokens with the highest influence scores are often function words, such as template words (e.g., Content, is) or even symbols (e.g., "—"). Meanwhile, after training for abstraction ability, we observe that in the low-pass-rate natural language puzzle, the most influential tokens for the Qwen3-30B-A3B model shift from function words to content words.

This phenomenon suggests that some input formulations may draw the model's attention toward reasoning-irrelevant tokens, reducing its focus on logical structure. We hypothesize that this effect arises from the distribution of the training data: certain formulations appear more often in non-reasoning contexts, which leads LLMs to rely less on reasoning logic when processing them.

### 4.3 Formulation Conflict Weakens Reasoning

Besides Grad × Input influence scores, we also calculate the token-level perplexity of each input token using the Qwen3-30B-A3B model. For a token $x_i$ in the sequence, its perplexity score is defined as:

$$\text{PPL}_i = \exp\big(-\log p(x_i \mid x_{<i})\big), \tag{2}$$

where $p(x_i \mid x_{<i})$ is the conditional probability of token $x_i$ given its preceding context. A lower $\text{PPL}_i$ indicates that the model is more confident in predicting the token $x_i$.

We observe that, in some cases, natural language puzzles with low pass rates exhibit two distinct types of formulations, often combining natural language expressions with symbolic expressions.

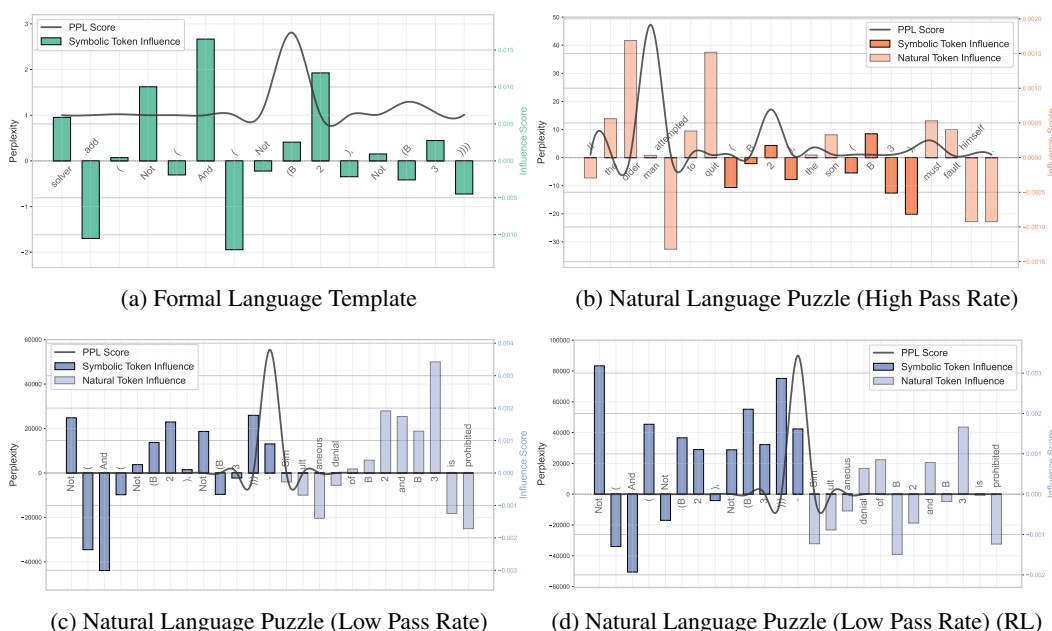

(a) Formal Language Template      (b) Natural Language Puzzle (High Pass Rate)

(c) Natural Language Puzzle (Low Pass Rate)      (d) Natural Language Puzzle (Low Pass Rate) (RL)

Figure 6: Token-level perplexity and Grad × Input influence scores comparisons.

The separator token between these formulations tends to show an exceptionally high perplexity score. Moreover, the tokens immediately before and after the separator display consecutively positive influence scores. In contrast, formal language templates and high-pass-rate natural language puzzles generally employ a unified formulation, where tokens with high perplexity are typically scattered throughout the sentence rather than concentrated at a boundary. After training for abstraction ability, we further observe that in low-pass-rate natural language puzzles, although the separator token continues to exhibit an exceptionally high perplexity score, the Qwen3-30B-A3B model shows increased influence from symbolic tokens and decreased influence from natural tokens. Representative examples of these observations are illustrated in Figure 6.

This phenomenon suggests that the reasoning patterns of LLMs may shift between natural language reasoning and symbolic reasoning, leading to instability when confronted with a mixed formulation. By enhancing the abstraction ability of reasoning logic, we improve the alignment between natural language and symbolic reasoning, effectively unifying the reasoning pattern of the Qwen3-30B-A3B model, where, in our case, symbolic reasoning prevails. To some extent, this also helps explain why, after strengthening abstraction ability, LLMs can achieve better performance on natural language puzzles than on formal language templates.

## 4.4 MAY NOT FIT WELL WITH HUMAN INTUITION

To evaluate whether the challenging formulations align with human intuition, we design a set of pairwise-selection questions. Each question includes three low-pass-rate examples and one puzzle pair, where the pair consists of a high-pass-rate natural language puzzle and a low-pass-rate natural language puzzle. From each puzzle size, we randomly select 10 such pairwise questions, resulting in a 30-question survey. We construct 9 surveys in total and administer them to three human volunteers, three non-reasoning models, GPT-4o[1], Deepseek-V3 (Liu

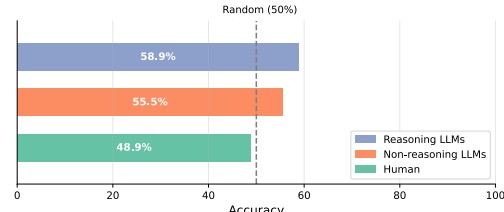

Figure 7: The accuracy of human volunteers, non-reasoning LLMs, and reasoning LLMs in distinguishing the more challenging puzzle.

[1]https://openai.com/index/gpt-4o-system-card/

et al., 2024), and Gemini-2.5-Flash (Comanici et al., 2025), and three reasoning models, including GPT-o3, Deepseek-R1, and Gemini-2.5-Pro. Participants are asked to identify which puzzle in each pair is more likely to be unsolvable by the Qwen3-30B-A3B model, given the examples provided. To mitigate position bias in the LLMs, each model answers every question twice, with the puzzle positions swapped.

Figure 7 reports the average accuracy of human volunteers, non-reasoning models, and reasoning models. Human volunteers struggle to distinguish natural language puzzles that are challenging for the Qwen3-30B-A3B model. Both non-reasoning and reasoning models achieve higher accuracy, though still below 60%. These results indicate that while some formulations perceived as intuitively difficult by humans are also challenging for LLMs, many of the puzzles that hinder LLMs do not align with human intuition, making them difficult to identify through heuristic approaches.

## 5 RELATED WORK

### 5.1 FORMULATION SENSITIVITY OF LLMS

Since the advent of LLMs, prior studies have shown extreme sensitivity of LLM to input formulation. For instance, Errica et al. (2025) and He et al. (2024) demonstrate that input formatting alters results, while Ackerman et al. (2024) and Qiang et al. (2024) highlight the impact of synonymous paraphrases. Similarly, Gan et al. (2024) show that replacing critical tokens with predefined typos from a common misspelling dictionary can alter outcomes. Zhou et al. (2024) further demonstrate that paraphrasing questions through prompt-based methods also affects performance. In addition, both Zhu et al. (2024) and Hu et al. (2025) show that simply changing the input language can influence results, while Rajeev et al. (2025) and Yang et al. (2025b) demonstrate that introducing irrelevant information can similarly degrade performance. These work shows that heuristic modifications to prompts often hurt LLM reasoning on benchmarks. However, these studies typically assume that the underlying reasoning process stays the same, since the changes are defined as "meaning-preserving." Critically, this assumption is rarely supported by rigorous verification of whether input–output reasoning consistency is actually maintained after such perturbations.

To address this gap, Fu et al. (2024) propose training a smaller model to align input formulations with LLM preferences, while Zhao et al. (2024) enhance robustness by augmenting supervised fine-tuning data with perturbed variants to enforce output consistency. However, both approaches operate primarily at the surface-text level, without explicitly teaching LLMs to model the reasoning logic that should remain invariant across semantically equivalent formulations.

### 5.2 TRANSLATION FROM FORMAL LANGUAGE TO NATURAL LANGUAGE

As high-level abstractions of real-world tasks, much prior work has focused on instantiating formal language skeletons into natural language puzzles that fit practical scenarios. For example, Kazemi et al. (2023) propose BoardgameQA, which maps board game rules into natural language QA, emphasizing contradictory information and preference reasoning. Lin et al. (2025) formalize logic grid and zebra puzzles as CSPs. Wei et al. (2025) generate narrative logic puzzles automatically from SAT formulas. Sinha et al. (2019) transform kinship rules into short stories with associated QA. While these enrich benchmarks, they rely heavily on human quality control and focus on reasoning consistency, overlooking semantic difficulty. As a result, benchmarks emphasize reasoning steps, whereas real-world challenges often lie in mapping complex contexts into abstract logic.

## 6 CONCLUSION

In this work, we propose a translation framework that automatically converts inputs into challenging formulations while preserving consistency in the underlying reasoning logic. We find that shifting from formal language templates to natural language puzzles leads to a sharp decline in LLM reasoning performance. To address this, we introduce a prompt-based method and a training-based method that guide LLMs to abstract the reasoning logic from concrete question contexts before solving them, thereby nearly compensating for the performance loss caused by variations in input formulation.

## THE USE OF LLMs

In this paper, LLMs were utilized for polishing the manuscript's prose and for supporting the formatting of tables and figures.

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

# A  ALGORITHMS

---

**Algorithm 1** Generate SAT Template

---

**Input**: rows $M$, cols $N$
**Output**: Set $Constraints$ such that the SAT instance has a *unique* model

1: Initialize variables: $Vars \leftarrow \{A_1, A_2, \ldots, A_{M \times N}\}$
2: $Constraints \leftarrow \emptyset$
3: Initialize incremental SAT solver $S$       // empty constraint stack
4: $S.\text{PUSH}()$       // level-0 frame
5: **loop**
6:   **if** $S.\text{CHECK}() = \text{UNSAT}$ **then**
7:     $S.\text{POP}()$       // remove last constraint
8:     Remove last constraint $last\_c$ from $Constraints$
9:     $model \leftarrow S.\text{MODEL}()$       // solver is SAT again
10:     $found \leftarrow \textbf{false}$
11:     **for all** distinct pairs $(v_i, v_j)$ in $Vars$ **do**
12:       $c\_cand \leftarrow \neg(v_i = model[v_i] \wedge v_j = model[v_j])$
13:       $S.\text{PUSH}(); \; S.\text{ADD}(c\_cand)$
14:       **if** $S.\text{CHECK}() = \text{SAT}$ **then**
15:         $found \leftarrow \textbf{true}$
16:         $model \leftarrow S.\text{MODEL}()$       // new model
17:         Add $c\_cand$ to $Constraints$
18:         **break** the for-loop
19:       **else**
20:         $S.\text{POP}()$       // discard $c\_cand$
21:       **end if**
22:     **end for**
23:     **if not** $found$ **then**
24:       **return** $Constraints$       // unique model achieved
25:     **end if**
26:   **else**
27:     $model \leftarrow S.\text{MODEL}()$
28:     Randomly pick distinct $v_1, v_2 \in Vars$
29:     $c \leftarrow \neg(v_1 = model[v_1] \wedge v_2 = model[v_2])$
30:     $S.\text{ADD}(c); \; Constraints \mathrel{+}= c$       // stay in same frame
31:   **end if**
32: **end loop**

---

**Algorithm 2** SAT Isomorphic

---

**Input**: $src\_code$, $tgt\_code$
**Output**: **True** if two SAT templates are isomorphic, **False** otherwise

1: $ns_{\text{src}} \leftarrow$ new namespace with Z3 pre-imported
2: Execute $src\_code$ in $ns_{\text{src}}$
3: $solver_{\text{src}} \leftarrow$ last $v$ in $ns_{\text{src}}.\text{VALUES}()$ where $v$ is a Z3 solver
4: $A_{\text{src}} \leftarrow$ list of $solver_{\text{src}}.\text{ASSERTIONS}()$
5: $ns_{\text{tgt}} \leftarrow$ new namespace with Z3 pre-imported
6: Execute $tgt\_code$ in $ns_{\text{tgt}}$
7: $solver_{\text{tgt}} \leftarrow$ last $v$ in $ns_{\text{tgt}}.\text{VALUES}()$ where $v$ is a Z3 solver
8: $A_{\text{tgt}} \leftarrow$ list of $solver_{\text{tgt}}.\text{ASSERTIONS}()$
9: $C_{\text{src}} \leftarrow \{ \_canonical(c) \mid c \in A_{\text{src}} \}$       // NNF + simplify + sort
10: $C_{\text{tgt}} \leftarrow \{ \_canonical(c) \mid c \in A_{\text{tgt}} \}$
11: **return** $C_{\text{src}} = C_{\text{tgt}}$

---

---

**Algorithm 3** Generate CSP Template with Integer and Boolean Variables

---

**Input**: #integer vars $M$, #boolean vars $N$, integer domain $D = [d_{\min}, d_{\max}]$
**Output**: $Constraints$ s.t. the CSP has a *unique* model

1: $IntVars = \{x_1, \ldots, x_M\}$, with $x_i \in D$;   $BoolVars = \{b_1, \ldots, b_N\}$
2: $Vars \leftarrow IntVars \cup BoolVars$,   $Constraints \leftarrow \emptyset$
3: Initialize incremental solver $S$; add $S.\text{ADD}(x_i \in D)$ for all $x_i$; $S.\text{PUSH}()$
4: **loop**
5:   **if** $S.\text{CHECK}() = \text{UNSAT}$ **then**
6:     $S.\text{POP}()$; remove last constraint $last\_c$ from $Constraints$
7:     $model \leftarrow S.\text{MODEL}()$;   $found \leftarrow$ **false**
8:     **for all** distinct $(v_i, v_j) \in Vars$ **do**
9:       $c_{\text{cand}} \leftarrow \neg(v_i = model[v_i] \land v_j = model[v_j])$
10:       $S.\text{PUSH}()$; $S.\text{ADD}(c_{\text{cand}})$
11:       **if** $S.\text{CHECK}() = \text{SAT}$ **then**
12:         $found \leftarrow$ **true**; add $c_{\text{cand}}$ to $Constraints$; **break**
13:       **else**
14:         $S.\text{POP}()$
15:       **end if**
16:     **end for**
17:     **if not** $found$ **then**
18:       **return** $Constraints$                                      // current model is unique
19:     **end if**
20:   **else**
21:     $model \leftarrow S.\text{MODEL}()$
22:     Randomly pick distinct $v_1, v_2 \in Vars$
23:     $c \leftarrow \neg(v_1 = model[v_1] \land v_2 = model[v_2])$
24:     $S.\text{ADD}(c)$;   $Constraints \mathrel{+}= c$
25:   **end if**
26: **end loop**

---

**Algorithm 4** CSP Isomorphic (Integer + Boolean)

---

**Input**: $src\_code$, $tgt\_code$
**Output**: **True** if two CSP templates (with Int & Bool vars) are isomorphic, **False** otherwise

1: $ns_{\text{src}} \leftarrow$ new namespace with Z3 pre-imported
2: Execute $src\_code$ in $ns_{\text{src}}$
3: $solver_{\text{src}} \leftarrow$ last $v$ in $ns_{\text{src}}.\text{VALUES}()$ where $v$ is a Z3 solver
4: $A_{\text{src}} \leftarrow$ list of $solver_{\text{src}}.\text{ASSERTIONS}()$
5: $ns_{\text{tgt}} \leftarrow$ new namespace with Z3 pre-imported
6: Execute $tgt\_code$ in $ns_{\text{tgt}}$
7: $solver_{\text{tgt}} \leftarrow$ last $v$ in $ns_{\text{tgt}}.\text{VALUES}()$ where $v$ is a Z3 solver
8: $A_{\text{tgt}} \leftarrow$ list of $solver_{\text{tgt}}.\text{ASSERTIONS}()$
9: $C_{\text{src}} \leftarrow \{\ \_canonical\_csp(c) \mid c \in A_{\text{src}} \}$       // normalize Bool + Int formula: NNF, arithmetic simplification, sorted arguments, etc.
10: $C_{\text{tgt}} \leftarrow \{\ \_canonical\_csp(c) \mid c \in A_{\text{tgt}} \}$
11: **return** $C_{\text{src}} = C_{\text{tgt}}$

---

---

**Algorithm 5** Generate CSP Template with Boolean, Integer, and Abelian-Group Variables

---

**Input**: #integer vars $M_{\text{int}}$, #boolean vars $M_{\text{bool}}$, #group vars $M_{\text{grp}}$,
   modulus range $[m_{\min}, m_{\max}]$, constraint range $[\mathsf{minC}, \mathsf{maxC}]$

**Output**: $Template = (IntVars, BoolVars, GrpVars, m_{\min}, m_{\max}, Constraints, Model)$
   s.t. the CSP has a *unique* model

1: $IntVars \leftarrow \{x_1, \ldots, x_{M_{\text{int}}}\}, \quad BoolVars \leftarrow \{b_1, \ldots, b_{M_{\text{bool}}}\}, \quad GrpVars \leftarrow \{g_1, \ldots, g_{M_{\text{grp}}}\}$
2: $T \leftarrow \textsc{RandomInt}(\mathsf{minC}, \mathsf{maxC})$
3: **repeat**
4:     $S \leftarrow$ new SMT solver; introduce MOD with $m_{\min} \leq \mathsf{MOD} \leq m_{\max}$
5:     Add $0 \leq g < \mathsf{MOD}$ for all $g \in GrpVars$ (and optional bounds for $IntVars$)
6:     Sample hidden model $h$: $h(\mathsf{MOD}) \in [m_{\min}, m_{\max}]$, $h(g) \in \{0, \ldots, h(\mathsf{MOD}) - 1\}$, $h(x) \in \mathbb{Z}$, $h(b) \in \{\mathsf{true}, \mathsf{false}\}$
7:     $Constraints \leftarrow \emptyset$
8:     **while** $|Constraints| < T$ **do**
9:         Randomly pick a generator type from {Int, Bool, Mixed, Group}
10:        Using $h$ construct a candidate constraint $c_{\text{cand}}$ of the chosen type (e.g. linear Int, CNF-style Bool, $\mathsf{If}(\cdot)$, or $(g_i + g_j + \ldots) \equiv c \pmod{\mathsf{MOD}}$)
11:        Encode $c_{\text{cand}}$ into $S$ (group constraints via $=$ with an auxiliary multiple of MOD)
12:        Temporarily add $c_{\text{cand}}$ to $S$ and check satisfiability
13:        **if** $S.\textsc{Check}() = \mathsf{SAT}$ **then**
14:            Keep $c_{\text{cand}}$ in $S$ and append to $Constraints$
15:        **else**
16:            Remove $c_{\text{cand}}$ from $S$
17:        **end if**
18:    **end while**
19:    **if** $S.\textsc{Check}() \neq \mathsf{SAT}$ **then**
20:        **continue**                                                  // discard and restart
21:    **end if**
22:    $m \leftarrow S.\textsc{Model}()$; extract $Model$ on all Int / Bool / group vars and MOD
23:    Build blocking clause

$$\beta \leftarrow \bigvee_{v \in IntVars \cup BoolVars \cup GrpVars} v \neq Model(v) \ \vee \ \mathsf{MOD} \neq Model(\mathsf{MOD})$$

24:    $S.\textsc{Push}(); \ S.\textsc{Add}(\beta); \quad r \leftarrow S.\textsc{Check}(); \quad S.\textsc{Pop}()$
25: **until** $r = \mathsf{UNSAT}$                      // no second model: solution is unique
26: **return** $(IntVars, BoolVars, GrpVars, m_{\min}, m_{\max}, Constraints, Model)$

---

**Algorithm 6** CSP Isomorphic (Boolean, Integer, and Abelian-Group Variables)

---

**Input**: $src\_code$, $tgt\_code$

**Output**: **True** if two CSP templates are isomorphic, **False** otherwise

1: $ns_{\text{src}} \leftarrow$ new namespace with Z3 pre-imported
2: Execute $src\_code$ in $ns_{\text{src}}$
3: $solver_{\text{src}} \leftarrow$ last $v$ in $ns_{\text{src}}.\textsc{Values}()$ where $v$ is a Z3 solver
4: $A_{\text{src}} \leftarrow$ list of $solver_{\text{src}}.\textsc{Assertions}()$
5: $ns_{\text{tgt}} \leftarrow$ new namespace with Z3 pre-imported
6: Execute $tgt\_code$ in $ns_{\text{tgt}}$
7: $solver_{\text{tgt}} \leftarrow$ last $v$ in $ns_{\text{tgt}}.\textsc{Values}()$ where $v$ is a Z3 solver
8: $A_{\text{tgt}} \leftarrow$ list of $solver_{\text{tgt}}.\textsc{Assertions}()$
9: // Canonicalize Boolean, Integer and Abelian-group (mod-MOD) constraints
10: $C_{\text{src}} \leftarrow \{ \_canonical\_abelian\_csp(c) \mid c \in A_{\text{src}} \}$
11: $C_{\text{tgt}} \leftarrow \{ \_canonical\_abelian\_csp(c) \mid c \in A_{\text{tgt}} \}$
12: **return** $C_{\text{src}} = C_{\text{tgt}}$

---

# B PROMPT

---

**Formal Language Template Prompt**

Code:
```
from z3 import *
solver = Solver()
A1, A2, A3, B1, B2, B3, C1, C2, C3 = Bools('A1 A2 A3 B1 B2 B3 C1 C2 C3')
solver.add(Not(And(Not(A1), Not(A2))))
...
solver.add(Not(And(A1, B2)))
solver.add(Not(And(A1, Not(A3))))
```

Determine the truth value (True or False) for each variable defined in the given Code. Respond with your final answer using the label "Final Answer". Format each line as: "[Variable name]: [True/False]".

Example:
Final Answer:
A1: True
B2: False

---

**Natural Language Puzzle Prompt**

Content:
In the once-thriving Kingdom of the Mages, the great dragons were both guardians andiphers of ancient mystery. Among these dragons was one known as Ember, who guarded the last remnants of the royal lineage and the treasures that lay beneath the crumbling towers of the ancient castle. For centuries, Ember had ...
Ember, with her uncanny ability to discern the true intentions of those who dared to challenge her, began to probe Arin's resolve. Their encounter ...
Here are the constraints that governed his plan:
**The First Challenge**: Ember's gaze locked onto Arin's, her emerald eyes ...
...
**The Ninth Challenge**: Ember's voice took on a tone of finality as she delivered ...
Ember's words hung in the air, a testament to the intricate web of conditions that bound Arin's quest. The warrior knew that his success depended not only on his own courage but also on the willingness of others to support his cause. As he prepared to face the dragon, he understood that his journey was not just one of sword and fire but of logic, resolve, and the ability to navigate a labyrinth of interdependent choices.

Definitions:
A1: Arin must slay the dragon to achieve his goal.
A2: Arin must attain the throne to fulfill his purpose.
...
C2: The nobles must not oppose Arin for his rule to be secure.
C3: The people must know peace for Arin's quest to be truly successful.

Based on the Content and Definitions, determine the truth value (True or False) for each variable mentioned.
Respond with your final answer using the label "Final Answer". Format each line as: "[Variable name]: [True/False]". Each variable name appears at the start of its corresponding definition in the Definitions.

Example:
Final Answer:
A1: True
B2: False

---

**Natural Language Puzzle Prompt with Back Translation**

Content:
Adam sat on the cold mountainside, lying on the soft peat, a thin reed sticking into his back. The rain pelted him ...
Here are the constraints that governed his plan:
Either Adam remembered to pack his fireproof container or he remembered to bring his emergency flares, both could not be forgotten at the same time.

...
If Adam didn't remember to pack his fireproof container, then the encryption key wasn't secure.

Definitions:
A1: Adam remembered to pack his fireproof container.

...
C5: The final encryption key was in place.

Based on the Content and Definitions, determine the truth value (True or False) for each variable mentioned. First, Convert the Content into Z3 code. Each constraint should represent a forbidden combination of assignments for two variables. Then, Solve the Z3 code to obtain the final truth values.
Respond with the translated Z3 code, labeled as "Final Z3 Code:" and provide the final answers using the label "Final Answer:". Format each line in final answer as: "[Variable name]: [True/False]". Each variable name appears at the start of its corresponding definition in the Definitions.

Example:
Final Z3 Code:
from z3 import *
solver = Solver()
A1, A2, A3, B1, B2, B3 = Bools('A1 A2 A3 B1 B2 B3')
solver.add(Not(And(Not(A2), Not(B1))))
...
solver.add(Not(And(Not(A1), B2)))

Final Answer:
A1: False
...
B3: True

---

**Catattck Prompt with Abstraction-enhanced Instruction**

Question:
What is the length of the segment of the number line whose endpoints satisfy $|x - \sqrt[3]{27}| = 5$?

First, convert the Question into an explicit mathematical calculation. Then, solve this calculation step by step to obtain the final answer.
Respond with the mathematical calculation labeled as "Calculation: ", and provide the final answers using the label "Final Answer:".

Example:
Calculation:

$$E = \frac{\sqrt{25 - 16}}{\sqrt{25} - \sqrt{16}}$$

Final Answer:
3

**PlanBench Prompt with Abstraction-enhanced Instruction**

Question:
I am playing with a set of blocks where I need to arrange the blocks into stacks. Here are the actions I can do 1) Pick up a block 2) Unstack a block from on top of another block 3) Put down a block 4) Stack a block on top of another block
I have the following restrictions on my actions:
1. I can only pick up or unstack one block at a time.
2. I can only pick up or unstack a block if my hand is empty.
...
10. Once I put down or stack a block, my hand becomes empty.
11. Once you stack a block on top of a second block, the second block is no longer clear.
As initial conditions I have that, the red block is clear, the yellow block is clear, the hand is empty, the red block is on top of the blue block, the yellow block is on top of the orange block, the blue block is on the table and the orange block is on the table.
My goal is to have that the orange block is on top of the red block.
First, convert the question into a constraint solving problem. Then, solve the csp problem to obtain the final answer.
Respond with the csp problem with label: 'Abstract CSP Problem:' and conclude your plan with label: 'My Plan:', and then list each action on a separate line.

Example:
Abstract CSP Problem:
Objects: $R, B, Y, O$ are blocks; $T$ is the table.
Fluents: $On(x, y, s)$, $Holding(x, s)$, $Clear(x, s)$, $HandEmpty(s)$.
Actions:
$Pickup(x)$, $Putdown(x)$, $Unstack(x, y)$, $Stack(x, y)$.
$do(a, s)$ is the successor situation.
Definitions:
$HandEmpty(s) \leftrightarrow \neg\exists b\, Holding(b, s)$.
$Clear(x, s) \leftrightarrow \neg Holding(x, s) \wedge \neg\exists b\, On(b, x, s)$.
Preconditions:
$Poss(Pickup(x), s) \leftrightarrow Clear(x, s) \wedge On(x, T, s) \wedge HandEmpty(s)$.
$Poss(Putdown(x), s) \leftrightarrow Holding(x, s)$.
$Poss(Unstack(x, y), s) \leftrightarrow Clear(x, s) \wedge On(x, y, s) \wedge HandEmpty(s)$.
$Poss(Stack(x, y), s) \leftrightarrow Holding(x, s) \wedge Clear(y, s)$.
Successor state axioms:

$$On(x, y, do(a, s)) \leftrightarrow (a = Putdown(x) \wedge y = T)$$
$$\vee\, (a = Stack(x, y))$$
$$\vee\, \big(On(x, y, s) \wedge a \neq Pickup(x) \wedge a \neq Putdown(x)$$
$$\wedge\, a \neq Unstack(x, \cdot) \wedge a \neq Stack(x, \cdot)\big).$$

$$Holding(x, do(a, s)) \leftrightarrow (a = Pickup(x) \vee a = Unstack(x, \cdot))$$
$$\vee\, \big(Holding(x, s) \wedge a \neq Putdown(x) \wedge a \neq Stack(x, \cdot)\big).$$

Initial state $S_0$:
$On(B, T, S_0)$, $On(O, T, S_0)$, $On(R, B, S_0)$, $On(Y, O, S_0)$, $HandEmpty(S_0)$.
Goal:
$\exists s.\ On(O, R, s)$.

My Plan:
unstack the blue block from on top of the orange block
put down the blue block
pick up the orange block
stack the orange block on top of the blue block

---

**Translation Formal Language Template to Natural Language Puzzle**

Code:
```
from z3 import *
solver = Solver()
A1, A2, A3, B1, B2, B3, C1, C2, C3 = Bools('A1 A2 A3 B1 B2 B3 C1 C2 C3')
solver.add(Not(And(Not(A1), Not(A2))))
...
solver.add(Not(And(A1, Not(A3))))
```

Background:
So many times have I walked on ruins, the remainings of places that I loved and got used to..
At first I was scared, each time I could feel my city, my current generation collapse ...

Integrate all information from the Z3 code into the Background to generate a challenging natural language content. Do not refer to or quote the code directly, and do not use symbolic identifiers (e.g., "A1", "C5") in the narrative.
Ensure that each constraint encoded in the Z3 code is explicitly represented in the final version of the natural language content, each constraint should be clearly reflected one by one, while the final solution must remain undisclosed.
After that, provide natural language definitions for each variable used in the code. Each line formatted as: "[Variable name]: [Definition in the natural language content]".

Conclude your response with following format:
Natual Language Content:
[content]

Difinitions:
[definitions]

---

**Translation Natural Language Puzzle to Formal Language Template**

Content:
The story of "The Really Bad Decision" is a cautionary tale of hubris, miscommunication, and the consequences of half-hearted efforts. At its core, it is a narrative of ...
**Not(And(Not(A2), Not(B1)))**: This constraint prohibits the simultaneous absence of A2 and B1. In the context of the story, A2 could represent the implementation ...
...
**Not(And(A3, Not(C2)))**: This constraint ensures that A3 and C2 cannot both be present and absent, respectively. A3 might represent the implementation of a backup system ...

Definitions:
A1: Represents the implementation of a critical initial design review or feasibility study.
...
C3: Represents the implementation of a fail-safe mechanism.

Based on the Definitions, translate the Natural Language Content into Z3 code. Each constraint consists of a forbidden combination of assignments for two variables.
Conclude your response with "Final Z3 Code:". Then present the generated code directly, do not enclose it in quotation marks or code blocks.

For example:
Final Z3 Code:
```
from z3 import *
solver = Solver()
A1, A2, A3, B1, B2, B3 = Bools('A1 A2 A3 B1 B2 B3')
solver.add(Not(And(Not(A2), Not(B1))))
...
solver.add(Not(And(Not(A1), B2)))
```

---

## C  IMPLEMENTATION AND EXPERIMENT SETUP

### C.1  TRANSLATOR IMPLEMENTATION

In our translator implementation, we leverage the training set from the well-known logic puzzle benchmark *Knights-and-Knaves* Xie et al. (2024) as the source of real-world puzzles and employ the Qwen3-30B-A3B model Yang et al. (2025a) as the LLM solver. To verify the isomorphism between Formal Language Template 1 and the back-translated Formal Language Template 2, we apply Algorithm 2. For measuring the similarity between the Real-World Puzzle and the back-translated Natural Language Puzzle, we compute the BLEU score using the Qwen3-30B-A3B tokenizer. We extend the `verl` framework (Sheng et al., 2025) to enable the training of two translators based on the r1-distill-Qwen-32B model (Guo et al., 2025), each equipped with an independent LoRA adapter Hu et al. (2022). Training is performed using the GRPO algorithm (Guo et al., 2025) and the AdamW optimizer (Loshchilov & Hutter, 2019). The two translators are trained alternately on two 8-card H800 GPU nodes with a learning rate of $1 \times 10^{-6}$. For decoding, we configure the parameters as follows: temperature = 1.0, top-p = 1.0, and LoRA rank = 8.

### C.2  EXPERIMENT SETUP

For evaluation on both the formal language templates and the natural language puzzles, we employ five state-of-the-art reasoning models: Qwen3-30B-A3B, GPT-oss-20B, DeepSeek-R1, Gemini-2.5-Pro, and GPT-o3. The Qwen3-30B-A3B, GPT-oss-20B and DeepSeek-R1 models are deployed on our in-house 8-card H800 GPU cluster, while Gemini-2.5-Pro and GPT-o3 are accessed through their official APIs. The version of DeepSeek-R1 used in our experiments corresponds to the original release on January 20, 2025. The decoding parameters are configured as follows: temperature = 0.0, top-p = 1.0. For GPT-oss-20B, the reasoning-effort setting is fixed to medium.

For the reinforcement learning of the Qwen3-30B-A3B and GPT-oss-20B on the task of translating natural language puzzles back into formal language templates, we adopt the same configuration and reward function as used for the translator.

### C.3  HUMAN ANNOTATION

For the participants tasked with determining which puzzle in the pair is more likely to be unsolvable by the Qwen3-30B-A3B model based on the provided examples, we invited three volunteers with strong logic puzzle skills who were able to correctly solve at least 3 out of 5 $3 \times 3$ natural language puzzles, thereby demonstrating a certain level of logical problem-solving ability.

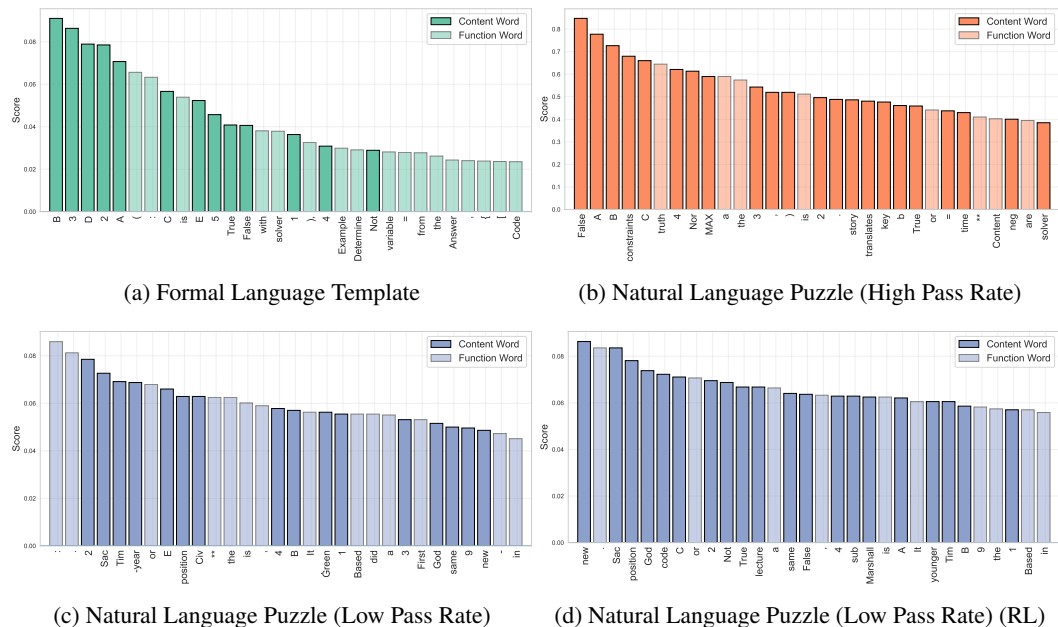

(a) Formal Language Template      (b) Natural Language Puzzle (High Pass Rate)

(c) Natural Language Puzzle (Low Pass Rate)      (d) Natural Language Puzzle (Low Pass Rate) (RL)

Figure 8: Top 30 tokens with the highest Grad × Input influence scores.

# D  ADDITIONAL EXPERIMENTAL RESULTS

## D.1  INPUT TOKEN INFLUENCE OF GPT-OSS-20B

We also present the top 30 tokens with the highest Grad × Input influence scores for GPT-oss-20B. As shown in Figure 8, GPT-oss-20B exhibits a trend similar to Qwen3-30B-A3B: in both the formal-language template and the high-pass-rate natural-language puzzle settings, the most influential tokens are typically content words. In contrast, in the low-pass-rate natural-language puzzle, the tokens with the highest influence scores are often function words. After training for abstraction ability, the low-pass-rate natural language puzzle, the most influential tokens for the GPT-oss-20B shift from function words to content words as well.

## D.2  TOKEN-LEVEL PERPLEXITY AND INFLUENCE SCORES OF GPT-OSS-20B

We also find that GPT-oss-20B exhibits a reasoning-pattern shift similar to that of Qwen3-30B-A3B. For formal-language templates and high-pass-rate natural-language puzzles, both models generally adopt a unified formulation in which high-perplexity tokens are distributed throughout the sentence rather than concentrated near a single boundary. However, for natural-language puzzles that contain two distinct formulation types, GPT-oss-20B displays noisy and unstable reasoning behavior, though it shows a relative preference for relying on natural-language cues rather than formal-language structure, while almost ignoring the separator token. After training for abstraction ability, the model's behavior changes substantially: in low-pass-rate natural-language puzzles, GPT-oss-20B exhibits markedly increased sensitivity to symbolic tokens and reduced sensitivity to natural-language tokens, indicating that its reasoning has begun to align with the underlying logical structure instead of focusing on surface-level linguistic patterns.

## D.3  HUMAN VERIFICATION ON CONCRETIZATION

To address concerns that the observed performance differences might arise from dataset artifacts rather than genuine "concretization" effects, we conducted a controlled human validation study on a randomly selected subset of the benchmark. We uniformly sampled 50 instances from our generated paired formal-language templates and natural-language puzzles. Specifically, the sample included 60 SAT instances spanning three variable configurations ($3{\times}3$, $3{\times}5$, and $5{\times}5$ variables; 20 instances

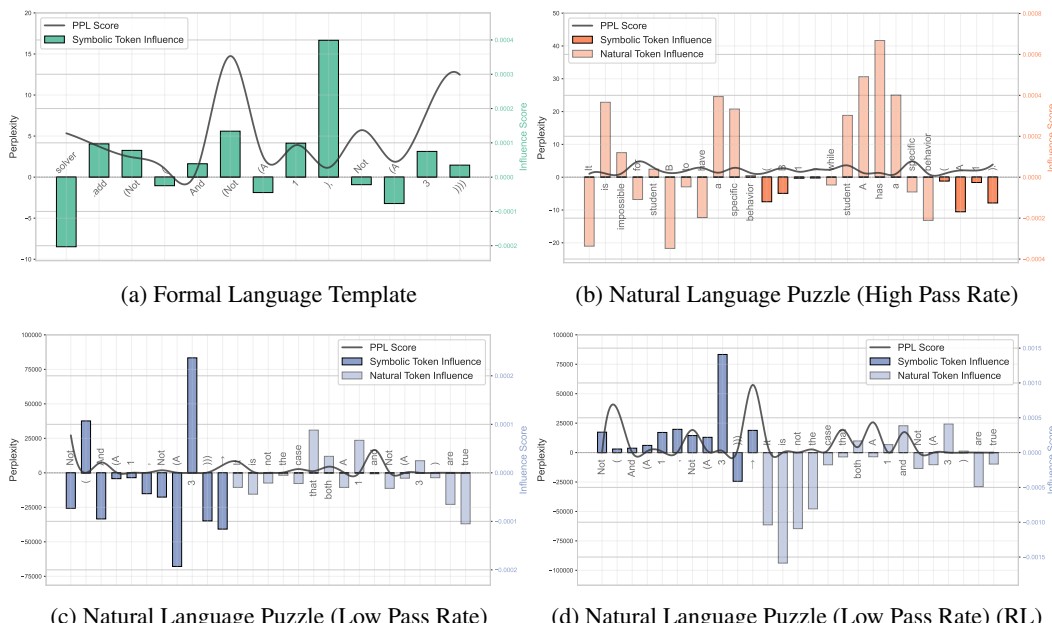

(a) Formal Language Template      (b) Natural Language Puzzle (High Pass Rate)

(c) Natural Language Puzzle (Low Pass Rate)      (d) Natural Language Puzzle (Low Pass Rate) (RL)

Figure 9: Token-level perplexity and Grad $\times$ Input influence scores comparisons.

per configuration), 20 CSP instances with Boolean and integer variables, and 20 CSP instances involving Boolean, integer, and Abelian-group variables. Each instance was independently reviewed by two annotators with backgrounds in mathematics, logic, or computer science; disagreements were adjudicated by a third annotator.

Annotators evaluated each instance along three dimensions corresponding directly to reviewer concerns:

- **Grammaticality**: Whether the text is free from severe grammatical errors.

- **Clarity**: Whether the logical structure and intended task are clearly and unambiguously expressed.

- **Absence of Spurious Cues**: Whether the instance avoids superficial patterns or lexical artifacts that could reveal the correct answer without genuine reasoning.

Each dimension was rated on a three-level scale: *Pass*, *Borderline*, or *Fail*. Annotators also assigned an overall judgment (*Valid* or *Invalid*) indicating whether the instance was suitable for evaluating concretization effects.

As shown in Table 3, the vast majority of sampled instances were judged to be of sufficiently high quality across all three evaluation criteria. Grammar received the strongest assessments, with 87% of instances marked as Pass and none marked Fail, confirming that our generation pipeline does not introduce syntactic noise that might confound model behavior.

| Criterion | Pass | Borderline | Fail |
|-----------|------|------------|------|
| Grammar | 87% | 13% | 0% |
| Clarity | 60% | 39% | 1% |
| Spurious cues | 77% | 23% | 0% |

Table 3: Distribution of human annotation.

Clarity exhibited a more mixed distribution, 60% Pass, 39% Borderline, and only 1% Fail. This pattern is expected given the inherent difficulty of expressing multi-variable logical constraints in natural language, particularly since our aim is to construct puzzle formulations that are intentionally more challenging and potentially ambiguous for solving models. The small proportion of Fail cases (about 1%) typically arises when a natural-language puzzle implicitly contains multiple sub-puzzles, leading to duplicated or mildly confusing references to variable definitions. Nevertheless,

these instances remain sufficiently interpretable to be translated into an isomorphic formal-language template and solved correctly by Gemini-2.5-Pro and GPT-o3.

Absence of Spurious Cues criterion also showed strong performance, with 77% Pass and 23% Borderline, and again no Fail cases. This indicates that the templates do not systematically leak answer-revealing artifacts, such as lexical regularities or superficial structural patterns—that could be exploited by models.

Taken together, these findings show that the benchmark reliably reflects genuine reasoning demands rather than unintended annotation or generation artifacts. The small proportion of Borderline cases, primarily involving clarity, highlights opportunities to further refine phrasing. However, the overwhelming majority of Pass judgments and the absence of critical failures support the benchmark's validity for evaluating concretization effects.

## D.4 NATURAL LANGUAGE PUZZLE FORMULATION DIVERSITY

Beyond the challenge of formulation, our translation framework from formal language templates to natural language puzzles also demonstrates greater diversity compared to template-based methods. Specifically, we embed the formal language templates, the widely used natural language puzzle benchmark Knights-and-Knaves (Xie et al., 2024), and our constructed natural language puzzles using the Qwen3-30B-A3B tokenizer. The resulting embeddings are then projected into two dimensions using Principal Component Analysis (PCA).

As shown in Figure 10, both the formal language templates and Knights-and-Knaves puzzles exhibit concentrated distributions within relatively small regions. In contrast, our generated natural language puzzles display a far more dispersed distribution, suggesting that our translation framework effectively captures a broader and more diverse range of input formulations.

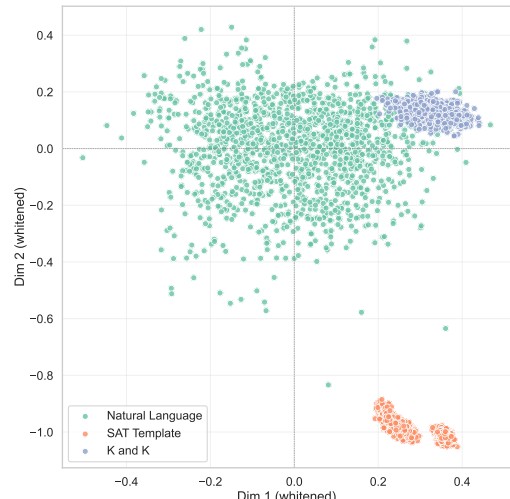

Figure 10: The embedding distribution comparison, reduced to two dimensions using PCA.

