# OpenReview forum: "Lost in Real-World Scenarios: Concretization Disrupts LLM Logical Reasoning"
_ICLR.cc/2026/Conference — Submitted to ICLR 2026_

### Official Review · Reviewer_2kAW · 2025-10-17

**Soundness:** 2
**Presentation:** 2
**Contribution:** 2
**Rating:** 2
**Confidence:** 3

**Summary:**

This paper propose a 2 translator system to understand the relationship between surface formulation and underlying reasoning logic during perturbations.

**Strengths:**

- The paper is well-motivated, the motivation question is important.

**Weaknesses:**

My overall comment of this paper is the motivation and method are a bit confusing. I cannot be fully convinced that the proposed 2-translator system can help answer the motivation question (line 49 -- 51).

- You've mentioned the gap is the community lacks deeper investigation into how LLMs model the relationship between surface formulation and underlying reasoning logic, how the concretization framework help bridge this gap? My understanding of the the framework is (1) translate FOL into NL and then (2) translated NL back to FOL. How this dual-translation process help review some patterns? In Line 44 -- 51,  perturbation methods are done in NL, but your starting point is FOL, will this be a mismatch?
- Please clarify if you think I misunderstand the overall structure. Translator 1 is from FOL to NL, translator 2 is from NL to FOL. You trained translator 1 using SAT problem, and trained translator 2 using real-world puzzles. What's the motivation of using these 2 translators? Are you going to discover some hidden reasoning structures during translation?
- Line 214, more training details are expected. What's the goal of the training.
- Figure 2 and figure 4 are a bit confusing. WHat's their relationships?

**Questions:**

- What Figure 1 is trying to convey? No reference for Figure 1. No explanations on the x-axis.
- Line 59 -- 66, can you explain a bit more about the shifting from formal templates to natural puzzles? My impression is, you have a dual-training process to train the model in order to have a good translation; whereas natural puzzles are OOD data, the gap is expected right?
- Sec 2.1 complexity analysis, how the analysis is related to the main argument of the paper, i.e., bridge the gap of udnerstanding discrepancies between surface formulation and underlying reasoning logic?
- Line 206, are you saying models perform bad in translating FOL to NL?

---

> ### Author Response · Authors · 2025-11-25
>
> Thank you very much for your thoughtful and constructive review of our work. We appreciate the time and effort you devoted to evaluating our submission. Below we address your questions and concerns in detail.
>
> 1. Role and Motivation of the Concretization Framework
>
> Your understanding of the dual-translator pipeline is correct: Translator 1 maps FL templates to NL puzzles, and Translator 2 maps NL puzzles back to FL templates.
>
> However, the dual-translator system does not function as a mechanism for inspecting the internal reasoning processes of LLMs. Instead, it primarily serves as a paired-data construction framework, providing an efficient mechanism for generating paired samples consisting of a formal language template and its logically equivalent natural language puzzle. This paired dataset serves two key purposes:
>
> - Abstraction–Concretization Analysis Data
>
> By evaluating non-trained LLMs on these paired instances, we observe a clear pattern: performance drops sharply once the same logic is expressed in concretized formulation. These observations reveal that the mapping from surface formulation to underlying reasoning logic is unstable for current LLMs and significantly limits their reasoning performance.
>
> - Abstraction-Enhanced Training Data
>
> Using the same paired data, we further apply our training-based, abstraction-enhanced method. We observe substantial improvements in performance on NL puzzles. These empirical results suggest that abstraction ability, the capacity to bridge the gap between surface formulations and their underlying logical structures, is essential for achieving robust and generalizable LLM reasoning.
>
> 2. Training Details of the Abstraction-Enhanced Method
>
> For the training-based abstraction-enhanced method, we train the solving model using a procedure similar to that of the NL→FL translator. The model takes as input a NL puzzle generated by our framework and is optimized through reinforcement learning. The reward signal depends solely on whether the model’s predicted formal language template is isomorphic to the original template. This objective explicitly encourages the model to ignore superficial linguistic variations and consistently map diverse concretized surface formulations back to their underlying abstract reasoning logic.
>
> 3. Relationship Between Figure 2 and Figure 4
>
> Figure 2 illustrates the construction pipeline for paired FL templates and NL puzzles, where both translators are employed to infer high-quality pairs. Figure 4, in contrast, depicts the dual-learning training procedure used to train the two translators themselves.
>
> 4. Explanation of Figure 1
>
> Figure 1 has been updated in the revised version. It presents the performance of Qwen3-30B-A3B on paired FL templates (Formal) and NL puzzles (Natural) across three task types, all constructed using our concretization framework. The figure also reports the model’s performance after applying our prompt-based (Prompt) and reinforcement-learning–based (Reinforced Learning) abstraction-enhanced methods.
>
> 5. Explanation of Line 59 -- 66: "Using the paired formal language templates and natural language puzzles constructed by our concretization framework, we observe a sharp reduction in LLM reasoning performance when moving from formal templates to natural puzzles, by 66% for the Qwen3-30B-A3B model.
>
> The LLM referenced in this sentence is not trained using the dual translators. The translators are employed solely to construct the paired abstraction–concretization dataset. We then evaluate LLMs (e.g., Qwen3-30B-A3B) on this paired data to measure their performance change when moving from the abstract formal statement to the concretized natural formulation.
>
> 6. Role of Sec 2.1 Complexity Analysis
>
> The complexity analysis serves as a supplementary explanation demonstrating the efficiency of our isomorphism verification procedure. We acknowledge that it is not directly connected to the central argument of the paper. In the revised version, we have replaced this section with a formal definition of our additional task: CSP problem with Boolean + integer variables and CSP problem with Boolean + integer + Abelian-group variables, in order to better illustrate the generality and applicability of our concretization framework.
>
> 7. Line 206: "To mitigate the reasoning performance gap of LLMs when transitioning from formal language templates to natural language puzzles"
>
> The sentence does not refer to the training of Translator 1 or Translator 2, and the LLM mentioned here is also not trained using our framework. Instead, it refers to the reasoning LLM, which achieves near-100% accuracy on the abstract FL templates but experiences a substantial performance drop when solving the corresponding NL puzzles.
>
> We have revised the inappropriate expressions in the updated manuscript. If any part of our response remains unclear or if additional details would be helpful, we would be happy to provide further clarification.

---

### Official Review · Reviewer_LTxT · 2025-10-25

**Soundness:** 3
**Presentation:** 3
**Contribution:** 3
**Rating:** 6
**Confidence:** 3

**Summary:**

The paper introduces a dual‑learning concretization pipeline that translates formal SAT templates into natural‑language puzzles while verifying isomorphism, letting the authors build paired items with matched logic. Across these pairs, state‑of‑the‑art models suffer large drops moving from formal language (FL) to natural language (NL). An abstraction‑first prompt mitigates the drop, and a training‑based (RL) variant that teaches the solver to back‑translate to templates recovers up to +56.2 points. Token‑level analyses implicate attention dispersion to non‑reasoning tokens and formulation conflicts.

**Strengths:**

* Brittleness to seemingly superficial phrasing is a core topic for reasoning LLMs; this work tackles it head‑on with paired, logic‑preserving FL↔NL instances and clear empirical evidence of fragility.
* Even a cutting‑edge reasoning model like Qwen3‑30B‑A3B collapses, sharply illustrating current limits of “reasoning” LLMs under realistic formulations.

**Weaknesses:**

* Dataset inclusion is governed by isomorphism verification and an LLM pass‑rate threshold, with no manual checks for grammaticality, clarity, unambiguity of variable definitions, or spurious cues. This raises the risk that some accuracy drops reflect dataset artifacts rather than purely “concretization” effects.

**Questions:**

* Did you run any manual check of samples to rate clarity, grammaticality, internal consistency, and unambiguity?,  to verify that the problems are reasonable for LLMs?
* The RL gains you report when training on data produced by your pipeline are intriguing. Do you expect the concretization/abstraction framework to extend beyond propositional SAT to first‑order (predicate) logic. Such an extension could plausibly generalize to a wider range of real‑world reasoning problems.

---

> ### Author Response · Authors · 2025-11-25
>
> Thank you very much for your thoughtful and constructive review of our work. We appreciate the time and effort you devoted to evaluating our submission. Below we address your questions and concerns in detail.
>
> 1. Manual Quality Check of Paired Abstraction–Concretization Data
>
> We uniformly sampled 100 items, including: 60 SAT instances across three variable configurations (20 each), 20 CSP instances with Boolean and integer variables, 20 CSP instances with Boolean, integer, and Abelian-group variables. Each item was independently reviewed by two annotators with computer science backgrounds and strong puzzle-solving skills; disagreements were resolved by a third annotator.
>
> Annotators evaluated each instance along the three dimensions raised by the reviewer: Grammaticality — Whether the text contains significant grammatical issues; Clarity — Whether the reasoning task is unambiguously and coherently expressed; Absence of Spurious Cues — Whether the instance avoids artifacts that offer shortcuts or reveal answers without genuine reasoning. Each criterion was scored as Pass / Borderline / Fail, and annotators also gave an overall Valid / Invalid judgment.
>
> |**Criterion**|**Pass**|**Borderline**|**Fail**|
> |---|---|---|---|
> |Grammar|87%|13%|0%|
> |Clarity|60%| 39%|1%|
> |Spurious cues|77%|23%|0%|
>
> The results show that over 98% of instances were judged Valid as evaluation items. Clarity has a relatively high proportion of Borderline cases, as expected given the complexity of expressing multi-variable constraints in natural language. The small proportion (about 1%) of Fail cases mainly arises when a natural-language puzzle implicitly contains multiple sub-puzzles, leading to duplicated or slightly confusing references to variable definitions. Nevertheless, these cases are still clear enough to be translated into an isomorphic formal-language template and solved correctly by Gemini-2.5-Pro and GPT-o3.
>
> 2. On Extending Concretization/Abstraction Beyond SAT
>
> Since SAT is a special case of constraint satisfaction problems (CSPs), where contains only Boolean variables, we expanded our evaluation to include more expressive CSP settings: Boolean + integer variables, Boolean + integer + Abelian-group variables.
> Across all these settings, we observed the same systematic FL→NL drop in state-of-the-art reasoning models, as well as consistent improvements with prompt-based abstraction and reinforcement-learning–based abstraction enhancement.
>
> | **Model** | **Method** | **Bool 3×3 FL** | **Bool 3×3 NL** | **Bool 3×5 FL** | **Bool 3×5 NL** | **Bool 5×5 FL** | **Bool 5×5 NL** | **+ Int FL** | **+ Int NL** | **+ Abel FL** | **+ Abel NL** |
> |---|---|---|---|---|---|---|---|---|---|---|---|
> |**Qwen3-30B-A3B**|Orig. | 97.6 | 31.6 | 89.4 | 29.8 | 41.4 | 23.2 | 99.4 | 36.4 | 90.6 | 62.6 |
> | |Prom. | - | 65.6 | - | 60.2 | - | 29.2 | - | 66.2 | - | 77.8 |
> | |RL| - | 87.8 | - | 84.4 | - | 53.4 | - | 80.2 | - | 83.2 |
> |**GPT-oss-20B** | Orig. | 85.8 | 74.2 | 62.0 | 49.2 | 20.2 | 13.0 | 97.8 | 70.2 | 84.4 | 47.8 |
> | |Prom.| - | 81.0 | - | 59.4 | - | 17.8 | - | 80.4 | - | 54.2 |
> | |RL|-| 86.4 | - | 74.2 | - | 24.2 | - | 83.8 | - | 72.8 |
> |**Deepseek-R1**|Orig.| 99.8 | 73.0 | 99.0 | 61.6 | 92.6 | 71.2 | 100 | 83.8 | 97.8 | 63.6 |
> | |Prom.| - | 92.2 |- | 81.0 | - | 76.4 | - | 88.4 | - | 71.8 |
> |**Gemini-2.5-Pro**| Orig. | 99.2 | 80.2 | 98.8 | 74.0 | 86.6 | 80.2 | 100 | 82.2 | 99.2 | 66.8 |
> | |Prom.|-|89.4|-| 78.8 | - | 68.6 | - | 87.4 | - | 76.0 |
> | **GPT-o3** | Orig. | 99.4 | 97.0 | 99.8 | 97.8 | 99.4 | 98.8 | 100 | 87.0 | 99.8 | 72.4 |
> | | Prom. | - | 98.0 | - | 99.6 | - | 99.0 | - | 90.4 | - | 83.6 |
>
> To test whether the abstraction skills acquired from our paired FL↔NL training data transfer to real-world reasoning tasks, we evaluated the RL-enhanced models on external benchmarks:
>
> | **Model** | **Method** | **Typographical Original** | **Typographical Edited** | **CatAttack Original** | **CatAttack Edited** | **Natural-Plan Calendar** | **Natural-Plan Meeting** | **Natural-Plan Trip** | **Planbench** |
> |---|---|---|---|---|---|---|---|---|---|
> | **Qwen3-30B-A3B** | Orig. | 90.47 | 86.87 | 96.50 | 94.50 | 84.80 | 12.30 | 3.75 | 68.20 |
> | | Prom. | 90.75 | 87.00 | 96.16 | 95.83 | 85.20 | 12.80 | 4.44 | 70.20 |
> | | RL | 91.18 | 87.50 | 96.50 | 96.00 | 86.20 | 14.10 | 4.94 | 73.20 |
> | **GPT-oss-20B** | Orig. | 79.18 | 70.81 | 63.00 | 61.00 | 83.90 | 4.00 | 0.00 | 47.40 |
> | | Prom. | 79.81 | 73.32 | 66.67 | 65.16 | 84.80 | 5.80 | 0.00 | 43.20 |
> | | RL | 82.41 | 76.11 | 70.60 | 70.00 | 85.70 | 9.60 | 0.06 | 55.60 |
>
> The RL-enhanced models consistently outperform baselines on the out-of-domain benchmarks, confirming that the learned abstraction behaviors extend beyond our synthetic setting to naturalistic planning and procedural reasoning tasks.
>
> We have incorporated all additional experimental results into the revised version. If any aspect of our response remains unclear or if further detail would be helpful, we would be happy to elaborate.

---

### Official Review · Reviewer_uoLB · 2025-10-27

**Soundness:** 4
**Presentation:** 3
**Contribution:** 3
**Rating:** 6
**Confidence:** 3

**Summary:**

This paper investigates the fragility of LLMs in logical reasoning when confronted with input formulation variations, moving beyond heuristic tests by proposing the concretization framework. This dual-learning framework automatically converts abstract Boolean satisfiability (SAT) formal language templates (FL) into challenging natural language puzzles (NL), using isomorphism verification to strictly ensure that the underlying reasoning logic remains consistent. Experiments using the resulting paired datasets revealed a sharp decline in LLM reasoning performance upon concretization. To address this, the authors proposed guiding LLMs to abstract the reasoning logic back into a formal template before solving (using prompt-based or training-based methods), successfully mitigating the performance loss.

**Strengths:**

1. The transformation framework is explained clearly with examples.
2. The proposed prompt-based method and the training-based method effectively mitigate the performance drop due to concretization formulation.
3. The analysis explains the possible cause of the performance decline, which are insightful.

**Weaknesses:**

1. The benchmark only investigates the problem with one task. More tasks are needed to prove the generalization of the framework.
2. The task seems not challenging enough, since GPT-o3 achieves >97% accuracy on all settings.
3. Only Qwen3-30B-A3B is used to validate the effectiveness of the prompting and RL-training framework. It would be better if you verify the effectiveness of the proposed methods with more models.

**Questions:**

1. It is more suitable to plot Figure 5 in bar chart instead of pie chart, since it is about absolute count instead of proportions.
2. Will you open-source the code and benchmark?

---

> ### Author Response · Authors · 2025-11-25
>
> Thank you very much for your thoughtful and constructive review of our work. We appreciate the time and effort you devoted to evaluating our submission. Below we address your questions and concerns in detail.
>
> 1. More Tasks to Demonstrate Generalization
>
> Beyond Boolean SAT, we have extended our concretization framework to mixed-domain CSPs involving Boolean + integer variables, and Boolean + integer + Abelian group variables. These settings substantially increase both computational and formulation complexity.
>
> Consistent with the SAT experiments, we observe a pronounced performance drop when moving from formal-language templates (FL) to natural-language puzzles (NL). Moreover, both our prompt-based and RL-based abstraction methods consistently mitigate this drop.
>
> The complete results are shown below:
>
> | **Model** | **Method** | **Bool 3×3 FL** | **Bool 3×3 NL** | **Bool 3×5 FL** | **Bool 3×5 NL** | **Bool 5×5 FL** | **Bool 5×5 NL** | **+ Int FL** | **+ Int NL** | **+ Abel FL** | **+ Abel NL** |
> |---|---|---|---|---|---|---|---|---|---|---|---|
> |**Qwen3-30B-A3B**|Orig. | 97.6 | 31.6 | 89.4 | 29.8 | 41.4 | 23.2 | 99.4 | 36.4 | 90.6 | 62.6 |
> | |Prom. | - | 65.6 | - | 60.2 | - | 29.2 | - | 66.2 | - | 77.8 |
> | |RL| - | 87.8 | - | 84.4 | - | 53.4 | - | 80.2 | - | 83.2 |
> |**GPT-oss-20B** | Orig. | 85.8 | 74.2 | 62.0 | 49.2 | 20.2 | 13.0 | 97.8 | 70.2 | 84.4 | 47.8 |
> | |Prom.| - | 81.0 | - | 59.4 | - | 17.8 | - | 80.4 | - | 54.2 |
> | |RL|-| 86.4 | - | 74.2 | - | 24.2 | - | 83.8 | - | 72.8 |
> |**Deepseek-R1**|Orig.| 99.8 | 73.0 | 99.0 | 61.6 | 92.6 | 71.2 | 100 | 83.8 | 97.8 | 63.6 |
> | |Prom.| - | 92.2 |- | 81.0 | - | 76.4 | - | 88.4 | - | 71.8 |
> |**Gemini-2.5-Pro**| Orig. | 99.2 | 80.2 | 98.8 | 74.0 | 86.6 | 80.2 | 100 | 82.2 | 99.2 | 66.8 |
> | |Prom.|-|89.4|-| 78.8 | - | 68.6 | - | 87.4 | - | 76.0 |
> | **GPT-o3** | Orig. | 99.4 | 97.0 | 99.8 | 97.8 | 99.4 | 98.8 | 100 | 87.0 | 99.8 | 72.4 |
> | | Prom. | - | 98.0 | - | 99.6 | - | 99.0 | - | 90.4 | - | 83.6 |
>
> 2. Is the Task Too Easy?
>
> In the extended mixed-domain CSP setting (Boolean + integer + Abelian group), where the pass rate of GPT-oss-120B is used as a difficulty threshold, even GPT-o3 reaches only 72% natural language accuracy, indicating that the task remains far from trivial. Nevertheless, GPT-o3 still benefits from our abstraction stage, achieving an 11.2% improvement.
>
> Our framework reveals two independent sources of difficulty: (1) the computational complexity of the underlying formal-language template, and (2) the formulation complexity introduced by natural-language concretization. The observed accuracy drop after concretization, paired with the accuracy gain after abstraction, demonstrates that abstraction guidance remains beneficial even for frontier reasoning models, helping mitigate brittleness introduced by surface-level linguistic variation.
>
> 3. Only Qwen3-30B-A3B is used to validate the effectiveness of the prompting and RL-training framework.
>
> We agree that validating across more models strengthens the claim. In addition to Qwen3-30B-A3B, we performed abstraction-enhancement experiments using GPT-oss-20B. The trend is strikingly consistent across models: both prompt-based and RL-based methods substantially reduce FL–NL performance gaps.
>
> Furthermore, GPT-oss-20B shows even stronger out-of-domain generalization under RL-based abstraction training, as demonstrated below.
>
> | **Model** | **Method** | **Typographical Original** | **Typographical Edited** | **CatAttack Original** | **CatAttack Edited** | **Natural-Plan Calendar** | **Natural-Plan Meeting** | **Natural-Plan Trip** | **Planbench** |
> |---|---|---|---|---|---|---|---|---|---|
> | **Qwen3-30B-A3B** | Orig. | 90.47 | 86.87 | 96.50 | 94.50 | 84.80 | 12.30 | 3.75 | 68.20 |
> | | Prom. | 90.75 | 87.00 | 96.16 | 95.83 | 85.20 | 12.80 | 4.44 | 70.20 |
> | | RL | 91.18 | 87.50 | 96.50 | 96.00 | 86.20 | 14.10 | 4.94 | 73.20 |
> | **GPT-oss-20B** | Orig. | 79.18 | 70.81 | 63.00 | 61.00 | 83.90 | 4.00 | 0.00 | 47.40 |
> | | Prom. | 79.81 | 73.32 | 66.67 | 65.16 | 84.80 | 5.80 | 0.00 | 43.20 |
> | | RL | 82.41 | 76.11 | 70.60 | 70.00 | 85.70 | 9.60 | 0.06 | 55.60 |
>
> Appendix D provides further analysis (e.g., Grad×Input influence scores, perplexity curves), showing that abstraction-enhanced training encourages GPT-oss-20B to rely less on superficial linguistic cues and more on underlying logical structure as well.
>
> We have incorporated all additional experimental results into the revised version. Figure 5 has been updated from a pie chart to a bar chart. We promise to open-source all code, datasets, and the full concretization framework upon acceptance. If any part of our reply remains unclear, we would be happy to provide further clarification.

---

### Official Review · Reviewer_Svik · 2025-11-01

**Soundness:** 2
**Presentation:** 2
**Contribution:** 3
**Rating:** 4
**Confidence:** 2

**Summary:**

Authors propose a concretization framework that translates reasoning logic to concrete context with challenging formulations. They use dual learning approach to train two translators with which paired datasets are generated. Their experiments show sharp decline LLM reasoning performance on this dataset?

**Strengths:**

1. New approach to generate data synthetically leading to a significant drop in performance is interesting and noteworthy.
2. The attention and perplexity analyses provide some insight into why performance may degrade under language variation.

**Weaknesses:**

1. The entire study is restricted to SAT-style logical reasoning, which is an extremely narrow slice of reasoning ability. It’s unclear if the findings or the proposed method to construct the dataset can extend to more representative "real-world” scenarios such as multihop reasoning or into other domains such as planning

3. The proposed prompt based approach is also very limited to SAT domains, for example authors include formal language template in the prompt, can the authors show that their method performs well on other domains and logical reasoning datasets as well?

4. Their experiments show that simple tweaking via prompts can boost the accuracy to almost 97% so there are some questions regarding the novelty of the work

**Questions:**

Please address the weakness

---

> ### Author Response · Authors · 2025-11-25
>
> Thank you very much for your thoughtful and constructive review of our work. We appreciate the time and effort you devoted to evaluating our submission. Below we address your questions and concerns in detail.
>
> 1. Study Limited to SAT-Style Logical Reasoning
>
> SAT is a classic problem in logical reasoning and serves as an abstract foundation for many real-world tasks, such as automated scheduling and constraint-based planning. Even state-of-the-art reasoning models exhibit a noticeable performance drop on our paired abstraction–concretization benchmark, indicating that SAT remains a valuable research target, one that combines meaningful reasoning difficulty and can be concretized into diverse natural-language problems with challenging surface formulations.
>
> Furthermore, we expanded our target problem to include more expressive CSP settings: Boolean + integer variables, Boolean + integer + Abelian-group variables. Consistent with the SAT experiments, we observe a pronounced performance drop when moving from FL templates to NL puzzles. Moreover, both our prompt-based and RL-based abstraction methods consistently mitigate this drop.
>
> The complete results are shown below:
>
> |**Model**|**Method**|**Bool 3×3 FL**|**Bool 3×3 NL**|**Bool 3×5 FL**|**Bool 3×5 NL**|**Bool 5×5 FL**|**Bool 5×5 NL**|**+ Int FL**| **+ Int NL**|**+ Abel FL**|**+ Abel NL** |
> |---|---|---|---|---|---|---|---|---|---|---|---|
> |**Qwen3-30B-A3B**|Orig.|97.6|31.6|89.4|29.8|41.4|23.2| 99.4|36.4|90.6|62.6|
> | |Prom.|-|65.6|-|60.2|-|29.2|-|66.2|-|77.8|
> | |RL|-|87.8|-|84.4|-|53.4|-|80.2|-|83.2|
> |**GPT-oss-20B**|Orig.|85.8|74.2|62.0|49.2|20.2|13.0|97.8|70.2|84.4|47.8|
> | |Prom.|-|81.0|-|59.4|-|17.8|-|80.4|-|54.2|
> | |RL|-|86.4|-|74.2|-|24.2|-|83.8|-|72.8|
> |**Deepseek-R1**|Orig.|99.8|73.0|99.0|61.6|92.6|71.2|100|83.8|97.8|63.6|
> | |Prom.|-|92.2|-|81.0|-|76.4|-|88.4|-|71.8|
> |**Gemini-2.5-Pro**|Orig.|99.2|80.2|98.8|74.0|86.6|80.2|100|82.2|99.2|66.8|
> | |Prom.|-|89.4|-|78.8|-|68.6 |-|87.4|-|76.0|
> |**GPT-o3** |Orig.|99.4|97.0|99.8|97.8|99.4| 98.8|100|87.0|99.8|72.4|
> | |Prom.|-|98.0|-|99.6|-|99.0|-|90.4|-|83.6|
>
> Beyond CSPs, abstraction-enhancement also improves reasoning in out-of-domain tasks, including planning benchmarks. This demonstrates the generalization ability of our proposed abstraction-enhancement approaches.
>
> |**Model**|**Method**|**Typographical Original**|**Typographical Edited**|**CatAttack Original**|**CatAttack Edited**|**Natural-Plan Calendar**|**Natural-Plan Meeting**|**Natural-Plan Trip**|**Planbench**|
> |---|---|---|---|---|---|---|---|---|---|
> |**Qwen3-30B-A3B**|Orig.|90.47|86.87| 96.50|94.50|84.80|12.30|3.75| 68.20|
> | | Prom.|90.75|87.00|96.16|95.83|85.20|12.80|4.44|70.20|
> | | RL |91.18|87.50|96.50|96.00|86.20|14.10|4.94|73.20|
> |**GPT-oss-20B**|Orig.|79.18|70.81|63.00| 61.00|83.90|4.00|0.00| 47.40 |
> | |Prom.|79.81|73.32|66.67|65.16|84.80|5.80|0.00|43.20|
> | |RL|82.41|76.11|70.60|70.00|85.70|9.60|0.06|55.60|
>
> 2. Prompt-Based Method is Limited to SAT Domain
>
> We agree that the effectiveness of the prompt-based method partly depends on providing accurate shots that capture the abstract structure of each task. As noted in the revision, we have included the full prompt templates used for CatAttack (primarily a mathematics benchmark) and PlanBench (mainly a Blocksworld planning benchmark) in Appendix B. This is precisely why we further propose the training-based abstraction-enhanced method, which does not require manually designed shots. Through training, the model internalizes abstraction ability, leading not only to improved performance on in-domain tasks but also to strong generalization to out-of-domain settings, including mathematics and planning tasks.
>
> 3. Large Boost from Prompting May Reduce the Novelty
>
> We are slightly puzzled by the reference to “97%,” as the only occurrence of this number in our paper corresponds to GPT-o3’s baseline (non-prompted) performance on the 3×3 and 3×5 SAT tasks. In fact, the most substantial improvements from prompting arise with Qwen3-30B-A3B: its natural-language accuracy improves from 31.6 → 65.6 (+34.0) with prompt-based abstraction enhancement.
>
> We believe the large boost from prompting does not diminish but rather underscores the novelty of our work. It serves as key evidence that the central challenge in real-world reasoning is not merely model capacity, but the lack of spontaneous abstraction over underlying logic, with models instead relying heavily on surface-level linguistic patterns. Furthermore, our RL-based abstraction training raises accuracy to 87.8 (+56.2 overall) and shows strong generalization performance. These results provide additional evidence that our abstraction-enhanced framework enables models to internalize abstraction capabilities, thereby improving robustness in real-world reasoning scenarios.
>
> We have incorporated all additional experimental results into the revised version. If any part of our reply remains unclear, we would be happy to provide further clarification.

---

> ### Comment · Reviewer_Svik · 2025-11-25
>
> Just for the purpose of clarification, I apologise if I have made any mistake, in the table 1 of the paper for 3x3 and 3x5 sat problems with gpt-o-3 the accuracy in NL tasks, the accuracy with the prompt based step is 98% right?

---

> ### Author Response · Authors · 2025-11-25
>
> Yes, that’s correct. Using the prompt-based method, the performance of GPT-o3 improves from approximately 97% to 98-99% on natural-language SAT problems with variable settings of 3×3 and 3×5. Let us know if the experimental results presented in Table 1 caused any confusion.

---

> > ### Comment · Reviewer_Svik · 2025-11-25
> >
> > Thanks for the clarification, For the open source models there is a noticeable increase in the performance with the RL based method, Would you be able to do a small abalation if the results of using external tools such as z3 solver results in a similar performance? If this is unfair or if the comparision is unnecessary can you clarify that?
> > I should have mentioned the above as a part of my questions earlier itself and I apologise for not doing that, I am willing to raise my score if you would be able to answer in my opinion this is an important result

---

> > > ### Author Response · Authors · 2025-11-25
> > >
> > > We are not entirely sure which component of the pipeline you would expect to ablate. For the formal language template, the accuracy of the Z3 solver is always 100%, since the ground-truth labels are generated directly by Z3. For the natural language puzzles, the accuracy of the Z3 solver is always 0%, because Z3 cannot process natural language inputs. Therefore, we assume that the pipeline you intend to compare is: the LLM first abstracts a natural language puzzle into a formal language template, and then Z3 solver is used as an external tool to solve this formal representation.
> > >
> > > We apologize for removing the Success Rate column from the revised Table 1 due to width constraints. Regarding Success Rate: we verify whether the formal language template generated by the LLM is isomorphic to the original template, including producing the same final answer. Thus, it can be interpreted as the accuracy obtained when Z3 solves the LLM-generated formal language template.
> > >
> > > | **Model** | **Method** | **3×3 FL** | **3×3 NL** | **3×3 Z3** | **3×5 FL** | **3×5 NL** | **3×5 Z3** | **5×5 FL** | **5×5 NL** | **5×5 Z3** |
> > > |-----------|------------|------------|------------|---------------|------------|------------|---------------|------------|------------|---------------|
> > > | **Qwen3-30B-A3B** | Orig. | 97.6 | 31.6 | - | 89.4 | 29.8 | - | 41.4 | 23.2 | - |
> > > | | Prom. | - | 65.6| 54.3 | - | 60.2| 66.3 | - | 29.2| 71.0 |
> > > | | RL | - || 90.1 | - | 84.4| 93.5 | - | 53.4| 88.9 |
> > >
> > > As shown in the results, on SAT-style tasks, the original Qwen3-30B-A3B model exhibits limited ability to abstract correct formal language templates. In particular, under the 3×3 setting, its performance when using Z3 is even worse than solving the problem end-to-end without external tools. After applying our RL-based method, the model’s abstraction ability is significantly improved: it generates more accurate formal representations, and consequently, its performance with Z3 as an external solver also increases.
> > >
> > > For the more challenging 5×5 setting, the reasoning capability of Qwen3-30B-A3B becomes the primary bottleneck, and Z3 serves as an effective complement. Incorporating Z3 as an external tool substantially improves overall reasoning performance. At the same time, this improvement also benefits from our RL-based method, which enables Qwen3-30B-A3B to generate more accurate formal language templates. These results indicate that, regardless of whether an external solver is used, the ability to correctly abstract the underlying logical structure is crucial for LLMs to learn and execute reasoning effectively.
> > >
> > > Finally, we emphasize the main motivation of our work: to study how LLM reasoning is affected when the same underlying abstract logic is expressed through concretized surface forms. Our findings show that formulation-induced interference is a major cause of the lack of robustness in LLM reasoning on real-world tasks, independent of whether external tools are integrated.

---

> ### Comment · Reviewer_Svik · 2025-11-25
>
> Most of my concerns have been adequately addressed by the authors hence I am raising my score. The rebuttals have also helped to understand the paper better therefore I have no strong objections to the paper being accepted.
>
> Wishing you all the best!

---

> > ### Author Response · Authors · 2025-11-25
> >
> > Thank you very much for your thoughtful feedback and for raising your score. We appreciate your recognition of our revisions and are glad that our responses have addressed your concerns. We sincerely appreciate your time and consideration.

---

### Author Response · Authors · 2025-11-29

Dear ACs,

Thank you for accepting our paper as a re-assigned submission. We truly appreciate the additional workload this may have caused. We are also grateful for the reviewers’ thoughtful and constructive feedback.

Although the early closure of the discussion period allowed us to engage only with Reviewer Svik, we are glad that the discussion reached a positive outcome, Reviewer Svik raised the score from 4 to 6 and the confidence from 2 to 3. At the same time, we regret that we were unable to discuss with the other reviewers, particularly Reviewer 2kAW, as we believe there may be a misunderstanding regarding the role and motivation of our proposed concretization framework.

Summary of the Reviews and Responses

We appreciate that the reviewers acknowledged the contributions of our work:

- Concretization framework, synthesizing paired abstract formal language templates and concretized natural language puzzles that share the same underlying reasoning logic. Reviewer Svik described it as “interesting and noteworthy,” and Reviewer uoLB stated that it “is explained clearly with examples.”

- Performance vary phenomenon, the degradation of LLM reasoning performance when transferring from formal language templates to natural language puzzles, and the performance mitigation achieved by our prompt-based and training-based abstraction-enhanced methods. Reviewer LTxT noted “clear empirical evidence of fragility” and “sharply illustrating current limits of reasoning LLMs under realistic formulations.”

- In-depth failure analysis, identifying root causes such as dispersed attention on non-reasoning tokens and alignment overhead between divergent reasoning patterns. Reviewer Svik commented that we “provide some insight,” and Reviewer uoLB considered our findings “insightful.”

The primary concern across the reviews centers on the generalization ability of our approach. Reviewer Svik noted that our setting focuses on “SAT-style logical reasoning, which is an extremely narrow slice of reasoning ability,” and commented that “the proposed prompt-based approach is also very limited to SAT domains.” Reviewer uoLB similarly emphasized that “more tasks are needed to prove the generalization of the framework” and that “it would be better if you verify the effectiveness of the proposed methods with more models.” Reviewer LTxT further asked, “Do you expect the concretization/abstraction framework to extend beyond propositional SAT to first-order (predicate) logic?”

In our response and revised manuscript, we strengthened evidence supporting generalization along three dimensions:

- Problem Types

Starting from SAT as a foundational reasoning domain underlying applications such as automated scheduling and constraint planning, we expanded evaluation to richer CSP settings (Boolean + integer variables; Boolean + integer + Abelian-group variables). Across these settings, we consistently observed substantial performance drops when moving from FL templates to NL puzzles, and both our prompt-based and RL-based abstraction-enhanced methods robustly mitigated these drops.

- Out-of-Domain Benchmarks

Beyond Typographical and CatAttack (math-focused benchmarks included in the initial version), we further evaluated on Natural-Plan and PlanBench (planning-focused settings). The prompt-based and training-based abstraction-enhanced methods consistently outperformed baselines on these out-of-domain benchmarks, demonstrating that learned abstraction behaviors transfer beyond synthetic settings to naturalistic planning and procedural reasoning tasks.

- Models Evaluated

Given the limited availability of open-source reasoning models when this work began, we newly evaluated GPT-oss-20B using both prompt-based and training-based methods on our paired abstraction–concretization datasets and out-of-domain benchmarks. Results remain strongly consistent: both approaches significantly reduce FL–NL gaps, and GPT-oss-20B exhibits even stronger out-of-domain generalization under RL-based abstraction-enhanced training.

These results collectively demonstrate that abstraction–concretization is a generally applicable principle rather than a SAT-specific technique, and that our proposed prompt-based and training-based abstraction-enhanced methods substantially improve robustness and out-of-domain generalization in real-world reasoning tasks.

For detailed experimental results and responses to reviewer-specific questions, please refer to our revised paper and our response to each reviewer. Thank you again for your time and effort.

Sincerely,

The Authors

---

### Meta-Review · Area_Chair_MesA · 2026-01-10

**Summary:**

The reject decision is primarily driven by concerns regarding this paper's scope and methodological clarity that remained unresolved. Doubts were raised about the validity of the dual-translator system, the motivation being confusing and the approach insufficient to convincingly decouple surface formulation from the underlying logic. While the authors attempted to address generalization concerns by expanding to other Constraint Satisfaction Problems (CSPs), the study remains heavily anchored in SAT-style logic puzzles. Concerns were also raised about the reliance on automated verification without rigorous manual quality control for the generated datasets. In its present form, the paper does not meet the bar for acceptance at ICLR.

**Reviewer Concerns:**

The rebuttal addressed Reviewer Svik's inquiries about the novelty of the findings compared to standard prompting and clarified the experimental data points that initially caused confusion. The authors made an effort to broaden the scope of the paper by adding Constraint Satisfaction Problems (CSPs) and evaluating additional models (GPT-oss-20B) in response to the feedback from Reviewers uoLB and Svik. However, significant concerns remain outstanding. Despite the new experiments, the study remains heavily anchored in SAT-style logic puzzles, failing to fully satisfy the demand for genuine "real-world" generalization raised by Reviewer uoLB. The concerns about the methodology raised by Reviewer 2kAW regarding the validity and motivation of the dual-translator system were not convincingly resolved, leaving the core premise of the framework in question. Finally, while the authors conducted a limited manual check in response to Reviewer LTxT, the reliance on predominantly automated verification without more rigorous quality control leaves open the risk that dataset artifacts influenced the results.

**Reviewer Scores:**

Only Reviewer Svik engaged with the rebuttal.
Below is the assessment for the remaining reviewers.
Reviewer uoLB would likely have maintained their positive rating. There is a small chance they might have increased it to a 7, if the authors satisfied their request for broader generalization by adding Constraint Satisfaction Problems (CSPs) and evaluating the additional GPT-oss-20B model. Similarly, there is a small chance that Reviewer LTxT might have raised their score to a 7, given that the authors conducted a manual quality audit on 100 items in order to address concerns about dataset artifacts. On the other hand, Reviewer 2kAW would likely have retained a low score (2 or 3), as their fundamental skepticism regarding the validity and motivation of the dual-translator system, specifically, whether it truly decouples surface formulation from logic, remained a core conceptual disagreement that the rebuttal's procedural clarifications were unlikely to fully resolve.

---

### Decision · Program_Chairs · 2026-01-26

Reject